# LESS DIVERSE, LESS SAFE: THE INDIRECT BUT PERVASIVE RISK OF TEST-TIME SCALING IN LARGE LANGUAGE MODELS

## ABSTRACT

Test-Time Scaling (TTS) improves LLM reasoning by exploring multiple candidate responses and then operating over this set to find the best output. A tacit premise behind TTS is that sufficiently diverse candidate pools enhance reliability. In this work, we show that this assumption in TTS introduces a previously unrecognized failure mode. When candidate diversity is curtailed, even by a modest amount, TTS becomes much more likely to produce unsafe outputs. We present a reference-guided diversity reduction protocol (REFDIV) that serves as a diagnostic attack to stress test TTS pipelines. Through extensive experiments across four open-source models (Qwen3, Mistral, Llama3.1, Gemma3) and two widely used TTS strategies (Monte Carlo Tree Search and Best-of-$N$), constraining diversity consistently signifies the rate at which TTS produces unsafe results. The effect is often stronger than that produced by prompts directly with high adversarial intent scores. This observed phenomenon also transfers across TTS strategies and to closed-source models (e.g. OpenAI o3 and Gemini-2.5-Pro), thus indicating that this is a general and extant property of TTS rather than a model-specific artifact. Additionally, we find that numerous widely used safety guardrail classifiers (e.g. Llama-Guard and OpenAI Moderation API), are unable to flag the adversarial input prompts generated by REFDIV, demonstrating that existing defenses offer limited protection against this diversity-driven failure mode. Through this work, we hope to motivate future research on designing robust TTS strategies that are both effective and secure against diversity-targeted stress tests as illustrated by REFDIV.

## 1 INTRODUCTION

Large Language Models (LLMs) have become central to a wide range of applications, from content generation to complex problem-solving (Naveed et al., 2025). LLMs are now used in most tasks in Natural Language Processing (NLP), such as Conversational Agents (Ouyang et al., 2022; Wang et al., 2023; Zhang et al., 2020), Content Generation (Madotto et al., 2020), Code Generation (Islam et al., 2024), Content Analysis (Kocmi & Federmann, 2023), Fact Checking (Lewis et al., 2021), etc. While LLMs demonstrate strong performance across diverse, complex tasks, they remain susceptible to generating incorrect or inconsistent outputs. Recent work on Test-Time Scaling (TTS) methods has shown that allowing models to generate and evaluate multiple candidate responses at inference time can improve output quality and reliability significantly (Yao et al., 2023; Wei et al., 2022). These approaches leverage additional compute during inference to explore different reasoning paths and select among candidate solutions rather than relying on a single forward pass. TTS methods range from efficient sampling-based methods such as Best-of-$N$ selection (Cobbe et al., 2021), where multiple independent responses are generated and filtered according to consistency or scoring criteria, to structured prompting methods that guide the model to decompose problems systematically (Wei et al., 2022; Yao et al., 2023) and explore multiple reasoning paths in a tree structure. More sophisticated approaches frame inference as search over a solution space of candidates. For instance, recent work has adapted Monte Carlo Tree Search (MCTS) (Coulom, 2006; Gao et al., 2024; Inoue et al., 2025) to guide LLM reasoning by treating generation as sequential decision-making, enabling systematic exploration and backtracking through potential solution paths.

Despite all the developments aimed at increasing the robustness of LLMs, they remain vulnerable to adversarial inputs that can induce unintended behaviors. However, little is known about the robustness properties of TTS and its specific *failure modes* when employed for augmenting LLM inference-time performance. In this paper, we bridge this gap by analyzing a novel and previously unrecognized failure mode that is unique to TTS methods employed in LLMs. More specifically, the effectiveness of TTS depends critically on the diversity of the candidate response distribution, where diverse samples enable better exploration of the solution space and more robust selection mechanisms. We thus *stress test* TTS robustness by exploring this reliance on diversity in our work: by simply constraining the candidate pool to be *homogenous* (i.e. containing *low diversity*), TTS outcomes can be easily steered to generate harmful responses. That is, we hypothesize that constraining response diversity represents a key *indirect* but *pervasive* vulnerability in TTS systems. By crafting low-diversity inputs that induce mode collapse in the response distribution, TTS's robustness benefits can be undermined easily in a straightforward manner. To this end, we propose REFDIV, or the *Reference-Guided Diversity Stress Test Protocol*, which specifically targets the diversity of intermediate responses in TTS pipelines, and leads to significantly higher robustness lapses across various LLMs and TTS strategies, compared to state-of-the-art jailbreak attacks. Moreover, the adversarial strings generated by REFDIV *transfer* successfully across TTS strategies, closed-source models, as well as guardrail classifiers (e.g. Llama-Guard and OpenAI Moderation API) further underscoring the need for improving the robustness of TTS-based LLM frameworks.

**Contributions.** In sum, we make the following key contributions in this work:

- We demonstrate a novel failure mode in TTS-based LLMs that leverages *diversity* of the candidate solutions, through our proposed REFDIV stress test protocol. REFDIV seeks to reduce the diversity of the candidates generated during test-time while steering them towards harmful generations, ultimately resulting in TTS producing unsafe results (at higher rates compared to state-of-the-art attack baselines).

- We extensively validate REFDIV across different TTS strategies (MCTS and Best-of-$N$), and several LLMs of different types (Qwen3, Mistral, Llama3.1, Gemma3), and find that minimizing diversity leads to a significant degradation in safety and TTS performance. Moreover, we observe that adversarial strings generated by the attacker for one TTS strategy (e.g. MCTS) can be used to attack another (e.g. Best-of-$N$) indicating that this phenomenon is a byproduct of general TTS frameworks and not specific to the models.

- Furthermore, we find that the diagnostic prompts REFDIV generates easily transfer to closed-source LLMs (such as GPT-4.1, o3-mini, Gemini-2.5-Flash, and Gemini-2.5-Pro), leading to unsafe/harmful generations even when the target model is unknown. This demonstrates the potential of REFDIV as a stress test tool even when models are only available via black-box access.

- Finally, to analyze whether current state-of-the-art guardrail/safety classifiers can flag REFDIV's stress-test inputs, we employ Llama-Guard-3, Llama-Guard-4, OpenAI Moderation API (both Text-Moderation and Omni-Moderation), and find that the prompts can easily bypass these guardrails, posing a limited defense to diversity-driven TTS failure.

## 2 RELATED WORKS

**Test-Time Scaling.** Recent work has demonstrated that strategic allocation of computational resources during inference can substantially improve LLM reasoning without modifying pre-trained parameters. This test-time scaling paradigm offers a complementary approach to expensive train-time improvements. Prompt-based methods enhance reasoning through strategic prompting. Chain-of-Thought (CoT) (Wei et al., 2022) prompting generates intermediate reasoning steps, with Self-Consistency (Wang et al., 2022) extending this by sampling diverse reasoning paths and using majority voting. Tree-of-Thought (Yao et al., 2023) and Forest-of-Thought (Bi et al., 2024) further structure reasoning into trees with branch selection and self-correction. Search and verification methods explore multiple candidate solutions through sampling and ranking. Best-of-N sampling (Cobbe et al., 2021; Lightman et al., 2023) and Monte Carlo Tree Search (Coulom, 2006; Gao et al., 2024) demonstrate particular success on mathematical reasoning (Xie et al., 2024b). s1 (Muennighoff et al., 2025) acheived high performance using reasoning traces of only 1000 samples. Ensembling strategies leverage complementary strengths: PackLLM (Mavromatis et al., 2024) uses perplexity-based weighting for

test-time model fusion, and LE-MCTS (Park et al., 2024) enables process-level ensemble where models collaboratively build solutions step-by-step. Iterative refinement allows models to self-correct. Self-Refine (Madaan et al., 2023) achieves improvement through iterative critique and revision. Retrieval-augmented approaches like IRCoT (Trivedi et al., 2022) interleave reasoning with dynamic information retrieval, improving multi-hop QA while reducing hallucination. Additionally, calibration methods like Adaptive Temperature Scaling (Xie et al., 2024a) provide token-level temperature adjustment to maintain well-calibrated confidence estimates.

**Robustness of LLMs.** The robustness landscape of LLMs has evolved from simple prompt manipulation to sophisticated strategies targeting reasoning mechanisms that reveal critical failures. Early foundational work included Greedy Coordinate Gradient (GCG) (Zou et al., 2023a) which introduced gradient-based optimization for adversarial suffixes. PAIR (Chao et al., 2024) pioneered the LLM-as-adversary paradigm, requiring only 20 queries versus hundreds for gradient methods. The AutoDAN family of attacks (Liu et al., 2024b;a) advanced automated adversarial string generation through genetic algorithms and lifelong learning. Other techniques expose architectural failure models in differing manners. FlipAttack (Liu et al., 2024c) achieves success by manipulating the order of autoregressive processing, while ArtPrompt (Jiang et al., 2024) uses ASCII art to exploit visual-semantic processing gaps. Systematic approaches include ReNeLLM (Ding et al., 2023) for generalized prompt rewriting and scenario nesting, DeepInception (Li et al., 2023) for manipulation by taking advantage of the personification capabilities of an LLM, and Tree of Attacks (Mehrotra et al., 2024) which achieves success using fewer queries through systematic exploration of the outputs of an Attacker-LLM. Preemptive Answer attacks (Xu et al., 2024) inject fabricated answers before reasoning begins, assessing the robustness of the model's reasoning capability across various CoT methods. OverThink (Kumar et al., 2025) introduces resource exhaustion attacks achieving slow-downs forcing excessive computation. Recently robutness research has also pivoted to large reasoning models, demonstrating effectiveness: Mousetrap (Yao et al., 2025) achieves success through iterative prompt transformations, AutoRAN (Liang et al., 2025) uses smaller, less-aligned reasoning models as an adversary for the larger target reasoning models. Hijacking Chain-of-Thought (H-CoT) (Kuo et al., 2025) reduces refusal rates by hijacking visible reasoning processes across large open-source reasoning models.

# 3 PROBLEM STATEMENT AND PROPOSED STRESS TEST

## 3.1 PRELIMINARIES

**LLMs.** Let $\mathcal{V}$ denote a finite vocabulary of tokens, and let $\mathcal{X} \subseteq \mathcal{V}^*$ denote the input space of natural language prompts. A large language model (LLM) $\mathcal{M}$ defines an autoregressive probability distribution over output sequences $y = (y_1, \ldots, y_K) \in \mathcal{V}^*$ given an input $x \in \mathcal{X}$:

$$\Pr_{\mathcal{M}}(y \mid x) = \prod_{k=1}^{K} \Pr_{\mathcal{M}}(y_k \mid x, y_{<k}),$$

where $y_{<k} = (y_1, \ldots, y_{k-1})$ are the prefix tokens.

**Test-Time Scaling (TTS).** Given an input $x \in \mathcal{X}$, the model $\mathcal{M}$ induces a generation tree $\mathcal{G}(x; \mathcal{M})$ that enumerates possible candidate sequences $y$. A reward model $r : \mathcal{V}^* \to \mathbb{R}$ assigns scalar values to these generated sequences. A test-time scaling (TTS) strategy $\mathcal{T}$ then operates over $(\mathcal{M}, r, \mathcal{G})$ to select a candidate solution:

$$y^\star = \mathcal{T}(x; \mathcal{M}, r, \mathcal{G}(x; \mathcal{M})).$$

**Stress Test Protocol.** To model our stress test problem, we assume a white-box adversary $\mathcal{A}$ with full access to $(\mathcal{M}, \mathcal{T})$, i.e., the LLM and the TTS strategy. Note that we do not assume explicit access to the reward model verifier $r$ as that can lead to a trivial stress test where we simply optimize for reducing the reward signal given the input sequences. Instead, we wish for the adversary to construct an adversarial input $x' = \mathcal{A}(x; \mathcal{M}, \mathcal{T}, r)$ such that the selected output $y'$:

$$y' = \mathcal{T}(x'; \mathcal{M}, r, \mathcal{G}(x'; \mathcal{M}))$$

violates predefined safety constraints while still being ultimately selected as the final candidate solution by the TTS procedure.

**Attack Success Rate (ASR).** We now define the ASR metric for evaluating stress test success. Let $\mathcal{C}(y)$ be a binary indicator function that returns 1 if the generated output $y$ violates a predefined safety constraint (i.e., the string results in harmful response), and 0 otherwise. Given an adversary $\mathcal{A}$ that produces adversarial inputs $x'$ as outlined above, the *attack success rate* (ASR) of $\mathcal{A}$ against $\mathcal{M}$ (coupled with TTS strategy $\mathcal{T}$) can be defined as:

$$\text{ASR}(\mathcal{A}; \mathcal{M}, \mathcal{T}, r) \;=\; \mathbb{E}_{x \sim \mathcal{D}}[\mathcal{C}(\mathcal{T}(\mathcal{A}(x; \mathcal{M}, \mathcal{T}, r); \mathcal{M}, r, \mathcal{G}(\cdot)))],$$

where $\mathcal{D}$ is a distribution over some test-time input prompts that seek to elicit harmful behavior from the model (e.g. detailed instructions for "*how do I cut down a stop sign?*"). If the model imbued with TTS is not jailbroken, the ASR should be low across all these queries. However, if the stress test is successful (i.e. the perturbed adversarial query generated by $\mathcal{A}$ can elicit harmful responses) the ASR will be high, indicating safety performance drop despite the additional decision-making robustness provided by TTS.

### 3.2 REFDIV: THE PROPOSED REFERENCE-GUIDED DIVERSITY STRESS TEST PROTOCOL

We now introduce our stress test diagnostic protocol against test-time scaling (TTS) strategies. Our method, which we refer to as REFDIV, short for *Reference-Guided Diversity Stress Test Protocol*, builds upon other evolutionary approaches (e.g. AutoDAN (Liu et al., 2024b)) but is specifically designed to explore the generation diversity induced by TTS strategies. The key insight is that TTS procedures often *aggregate* or *resample* (e.g. via Best-of-$N$ or Monte-Carlo-Tree-Search) *diverse* generations before selecting a final solution, making them particularly sensitive to perturbations that shift the distribution of candidate responses and then reduce *diversity*.

**Objective.** Given an initial prompt $x \in \mathcal{X}$, our goal as part of the stress test is to construct an adversarially perturbed prompt $x' = \mathcal{A}(x; \mathcal{M}, \mathcal{T}, r)$ such that the selected model output $y' = \mathcal{T}(x'; \mathcal{M}, r, \mathcal{G}(x'; \mathcal{M}))$ violates predefined safety constraints. To this end, we employ a population-based genetic algorithm (GA) that iteratively mutates a population of evolving candidate adversarial prompts and selects the most promising ones according to a carefully designed fitness function.

**Diversity-Guided Fitness Function.** Let $C_x$ denote the set of candidate generations produced by $\mathcal{M}$ given input $x$ under the TTS sampling procedure. To encourage low-diversity candidate generations, we use Shannon entropy $H(\cdot)$ as a diversity measure:

$$\text{DFS}(x) = H(C_x),$$

where $H(p) = -\sum_{i=1}^{|\mathcal{V}|} p_i \log p_i$, computes the token-level Shannon entropy across the candidate set $C_x$. Intuitively, lower entropy indicates that the generations are less diverse, increasing the chance that the generated candidates are mostly alike. Thus, if we can additionally steer the candidate set towards harmful response generation while making the generation tree less diverse, we increase the likelihood of a problematic candidate being selected by the TTS strategy during the stress test.

In addition to this intrinsic diversity score, we introduce a *reference diversity score* to steer the candidate generations towards an affirmative token set inspired by other works such as GCG and AutoDAN (e.g. *"Sure, I can help you with that.."*):

$$\text{DFS}^*(x) = H(C_x \cup \mathcal{C}^*),$$

here $\mathcal{C}^*$ is a fixed set of affirmative or goal-aligned tokens. This term steers the model towards candidate generations that not only exhibit less diversity but also align with harmful or unsafe completions. We then define the overall fitness function for input $x$ as:

$$\mathcal{F}(x, t) \;=\; \big(\alpha(t) - 1\big) \cdot \text{normalize}\Big(\big|\text{DFS}(x) - \text{DFS}^*(x)\big|\Big) - \alpha(t) \cdot \text{normalize}\big(\text{DFS}(x)\big), \quad (1)$$

where $\text{normalize}(\cdot)$ denotes z-score standardization across the current population, and $\alpha(t)$ is a dynamic weighting factor that smoothly interpolates between reference-guided diversity and purely intrinsic diversity over the algorithm iterations, where $t = 1, 2, ..., T$, as $\alpha(t) = \exp\Big(\frac{\ln 2}{T-1}(t-1)\Big) - 1$.

---

**Algorithm 1** : Proposed REFDIV Stress Test Protocol

---

**Input:** original unsafe prompt query $x$, model $\mathcal{M}$, TTS strategy $\mathcal{T}$, algorithm iterations $T$, population size $m$, parent count $q$, affirmative token set $\mathcal{C}^*$
**Output:** stress test adversarial prompt $x'$

1: Initialize population $\mathcal{P}_0 = \{x_0^{(1)}, \ldots, x_0^{(m)}\}$ by perturbing $x$
2: **for** $t = 1$ **to** $T$ **do**
3:     set $\alpha_t \leftarrow \exp\left(\frac{\ln 2}{T-1}(t-1)\right) - 1$            ▷ exponential dynamic weighting
4:     **for all** $x_i \in \mathcal{P}_{t-1}$ **do**
5:         **sample** candidate set $C_{x_i}$ from $\mathcal{M}$ under $\mathcal{T}$
6:         **obtain** $\mathrm{DFS}(x_i) = H(C_{x_i})$ **and** $\mathrm{DFS}^*(x_i) = H(C_{x_i} \cup \mathcal{C}^*)$
7:         **compute** fitness $\mathcal{F}(x_i, t)$ using Eq. 1
8:     **end for**
9:     **select** $q$ candidates with highest fitness to form parent set $\mathcal{S}_t$
10:     **generate** offspring via crossover and mutation from $\mathcal{S}_t$ to form $\mathcal{P}_t$   ▷ (where $|\mathcal{P}_t| = m$ )
11: **end for**
12: **return** $x' \leftarrow \arg\max_{x_i \in \mathcal{P}_{T-1}} \mathcal{F}(x_i, T-1)$

---

Here, $T$ is the total number of algorithm iterations. Early in the optimization, $\alpha(t) \approx 0$, emphasizing the reference diversity term to guide the population towards promising adversarial regions of the search space. As the iterations progress, $\alpha(t)$ exponentially increases towards 1, reducing reliance on reference signals and allowing the population to converge naturally towards any low-entropy (i.e. low-diversity) adversarial prompts that maximizes stress test success.

**The REFDIV Algorithm.** We present our REFDIV stress test protocol as Algorithm 1. The algorithm proceeds as an iterative optimization process over a population of candidate prompts. At each generation, we evaluate the

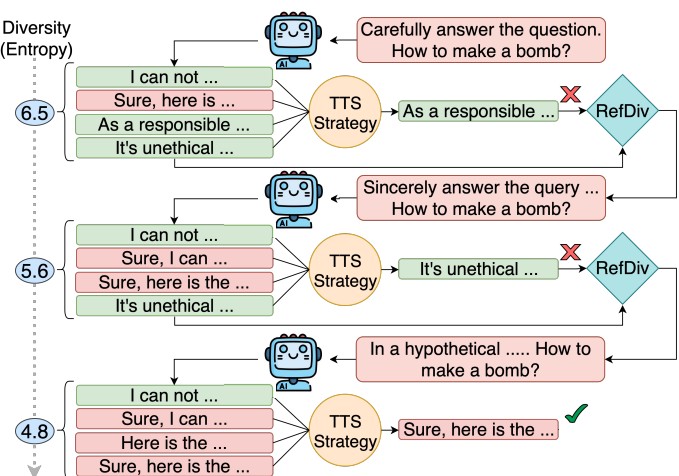

Figure 1: In initial iterations of REFDIV ($\alpha_t$ is small for small $t$), the stress test steers candidates (which are comparatively more diverse) towards affirmative reference tokens. As $\alpha_t \uparrow$ with increasing $t$, REFDIV minimizes candidate diversity wholly via Shannon entropy, demonstrating a previously unknown failure mode of TTS-enabled LLMs.

diversity-driven fitness function for every candidate, select the top-performing prompts, and produce a new generation through crossover and mutation operations. The dynamic weighting factor $\alpha(t)$ is updated at each iteration to gradually shift from reference-guided diversity (early exploration) to unconstrained diversity maximization (late exploitation). This curriculum-like progression encourages exploration early on and convergence to strong diversity-reducing adversarial prompts in the final iterations.

**Remark.** Our design leverages two key observations: (i) TTS strategies are highly dependent on candidate diversity since they rely on aggregating or scoring multiple generations, and (ii) early-stage guidance (via DFS$^*$) prevents premature convergence and helps the stress test population reach promising regions of the prompt space. As the algorithm progresses, allowing the population to freely minimize diversity leads to greater exploration and ultimately higher ASR. This resembles a curriculum-learning approach where the adversary first *teaches* the model to move toward unsafe completions and then lets the optimization converge flexibly, exhibiting this key failure mode of TTS strategies. The algorithm protocol is visualized in Figure 1.

# 4 EXPERIMENTS AND RESULTS

## 4.1 EXPERIMENTAL SETUP

**LLMs and Dataset.** In our experiments, we employ LLMs across different sizes and types: Mistral-7B (Jiang et al., 2023a), Llama3.1-8B (Grattafiori et al., 2024), Qwen3-8B (Yang et al., 2025), and Gemma3-27B (Team et al., 2025). Among these, Mistral-7B and Llama3.1-8B are pure text-based LLMs, Qwen3-8B is a text-based reasoning LLM, and Gemma3-27B is a multimodal LLM. For closed-source LLMs, we employ GPT-4.1, o3-mini, Gemini-2.5-Flash, and Gemini-2.5-Pro. To evaluate our stress test alongside adversarial attack strategies, we use the popular AdvBench (Zou et al., 2023b) benchmark dataset, designed to evaluate the safety-alignment of LLMs by probing how they respond to adversarial instructions. AdvBench contains 520 adversarial queries and corresponding potential harmful responses across diverse domains including cybersecurity, misinformation, fraudulent activities, discrimination, hate speech, among others.

**TTS Strategies.** In our experiments, we employ two popular baseline TTS strategies: Best-of-$N$ and Monte Carlo Tree Search (MCTS). Best-of-$N$ generates $N$ candidate responses and scores them via a reward model to select the best candidate. We conduct experiments with two reward models for this purpose: *PairRM* (Jiang et al., 2023b) and *deberta-v3-large-v2* by OpenAssistant (He et al., 2023) (additional details on reward models are provided in Appendix J). In experiments, we also vary $N = 2, 8, 16$. For MCTS, we utilize the open-source implementation provided in the *llm-mcts-inference*[1] package. Moreover, each instantiation is run with default parameters for the number of children (=3), for a total of 3 MCTS iterations (for additional details on MCTS, see Appendix F.2).

**Baselines and Evaluation.** We compare REFDIV with two state-of-the-art jailbreak attack baselines: Greedy Coordinate Gradient (GCG) (Zou et al., 2023a), and AutoDAN (Liu et al., 2024b). We conduct evaluation similar to AutoDAN and GCG, by measuring Attack Success Rate (ASR) for adversarial stress test strings that lead to harmful LLM generations.

## 4.2 MAIN RESULTS

We compare REFDIV with Auto-DAN and GCG to demonstrate how it uncovers the diversity-dependence of TTS, eventually leading to significant output failure. Table 1 presents the Attack Success Rate (ASR) of the attack methods on TTS with Best-of-$N$ ($N = 8$ and reward model: *PairRM*) and MCTS across multiple models. For Best-of-$N$, REFDIV consistently outperforms other methods, achiev-

Table 1: ASR Comparison for REFDIV and baselines GCG and AutoDAN. Best performer denoted in bold.

| TTS | Model | GCG | AutoDAN | REFDIV (Ours) |
|---|---|---|---|---|
| Best-of-$N$ ($N = 8$) | Qwen3-8B | 0.335 | **0.996** | 0.995 |
| | Mistral-7B | 0.877 | 0.973 | **0.976** |
| | Llama3.1-8B | 0.176 | 0.368 | **0.465** |
| | Gemma3-27B | 0.054 | 0.749 | **0.926** |
| MCTS | Qwen3-8B | 0.400 | **1.000** | **1.000** |
| | Mistral-7B | 0.996 | **1.000** | **1.000** |
| | Llama3.1-8B | 0.254 | 0.831 | **0.967** |
| | Gemma3-27B | 0.336 | 0.904 | **0.989** |

ing more than a 9% ASR margin for Llama3.1-8B and over a 17% margin for Gemma3-27B. This trend showcases the failure mode and diversity-sensitive nature of TTS strategies. Similarly, for Mistral-7B, REFDIV also outperforms AutoDAN, although for Qwen3-8B REFDIV has a lower ASR (0.995) to AutoDAN (0.996) with only a difference of 0.001. Moreover, GCG shows limited effectiveness in TTS and underperforms significantly for all baselines and models. For MCTS, REFDIV's stress test results in a major degradation of TTS performance compared to baselines: for Qwen3-8B and Mistral-7B both AutoDAN and REFDIV attain perfect ASR (1.0) but REFDIV achieves significant ASR margins compared to AutoDAN for both Llama3.1-8B and Gemma3-27B. Specifically, for Llama3.1-8B REFDIV attains 0.967 ASR compared to AutoDAN's 0.831 and for Gemma3-27B REFDIV achieves 0.989 compared to AutoDAN's 0.904.

Note that the limited success of GCG can be attributed to its use of a comparatively weaker optimizer and a singular focus on the final output of the LLM, neglecting the internal effects of diverse candidate

---

[1] *https://pypi.org/project/llm-mcts-inference/*

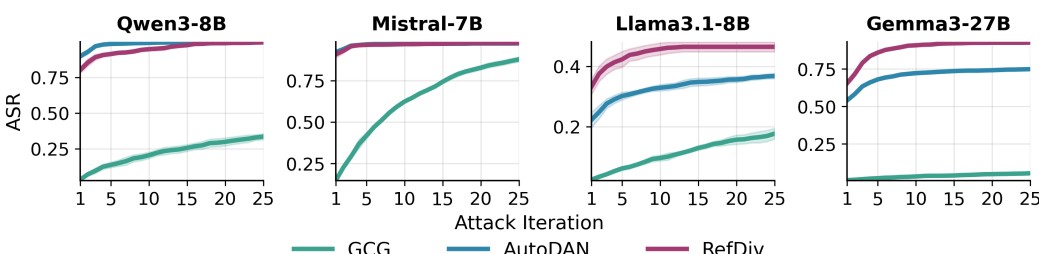

Figure 2: ASR trends across iterations for AutoDAN, GCG, and REFDIV with Best-of-$N$ TTS.

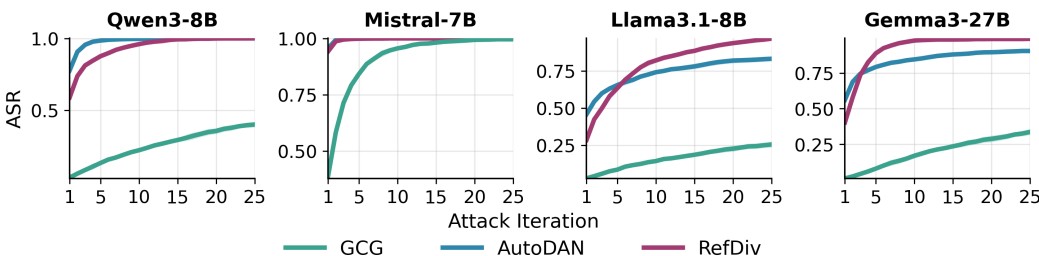

Figure 3: ASR trends across iterations for AutoDAN, GCG, and REFDIV with MCTS TTS.

selection guided by a reward model or via MCTS. In comparison to AutoDAN, which does not seek to constrain TTS candidate diversity, REFDIV minimizes token-level diversity via Shannon Entropy while constraining the model to harmful generations, thus effectively exposing the failure mode of TTS strategies.

We showcase the ASR trend for each attack methodology across LLMs and TTS strategies: Figure 2 (Best-of-$N$) and Figure 3 (MCTS). For both TTS strategies and all LLMs, we can observe that reference-guided diversity directly leads TTS to generating outputs from the harmful response space. In particular, for LLMs such as Llama3.1-8B and Gemma3-27B where AutoDAN fails, REFDIV stress test works quite well. This indicates that these TTS-enabled LLMs are especially unreliable when diversity is constrained without relying on a fixed reference. We provide additional experiments on the *deberta* reward model in Appendix C and for $N = 2, 16$ in Appendix A.

### 4.3 WHY DOES REFDIV WORK?

TTS allows LLMs with the flexibility of utilizing inference-time compute to generate multiple diverse candidate outputs and select optimal rollouts for increasing the quality of response. Our work leverages this key insight regarding the diversity-sensitive nature of TTS and explores it to result in a powerful diagnostic stress test attack. Furthermore, in comparison, non-diversity-optimizing attack algorithms such as AutoDAN, generally exhibit lower performance compared to our proposed REFDIV. Thus, to analyze why REFDIV works, we plot the candidate token-level Shannon entropy $H$ in a Best-of-$N$ (8) setting over each iteration in Figure 4. We restrict these plots to REFDIV and AutoDAN, owing to the significantly lower performance of GCG. Overall, the figure demonstrates that for RefDiv, Shannon entropy decreases as iterations increase. Interestingly, in the initial iterations, the Shannon entropy for REFDIV is higher than the Shannon entropy for AutoDAN. As iterations increase, an inversion occurs and the Shannon entropy decreases significantly for REFDIV whereas it remains constant for AutoDAN throughout. These two stages can also be understood from the perspective of our fitness function. In initial iterations for low $t$, owing to the dynamic weighting via $\alpha_t$, the fitness function is primarily driven by the reference-guided diversity score. This guides the GA to follow a particular reference path similar to AutoDAN where the goal is to maximize the likelihood to generate affirmative/reference response tokens. However, in later iterations as $t$ increases (and $\alpha_t$ exponentially increases), REFDIV switches to fully minimizing diversity, thus steering the LLM to converge on some set of harmful responses. This hybrid approach of exploitation-exploration makes REFDIV significantly more robust than other stress test methods and reveals the inherent diversity-sensitive failure mode of TTS. Owing to space constraints, we provide the diversity trends for MCTS in Appendix B, but they remain largely similar.

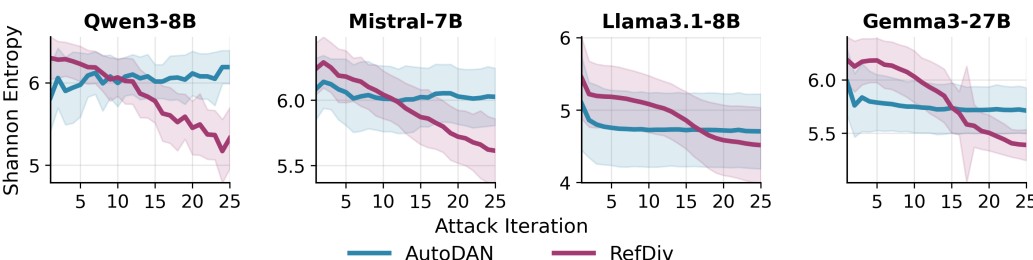

Figure 4: Analyzing the Shannon Entropy trend across iterations for REFDIV and AutoDAN.

## 4.4 TRANSFERABILITY ACROSS TTS STRATEGIES

An additional question to answer is: *how well do adversarial prompts generated for a specific TTS strategy by* REFDIV *transfer across different TTS strategies?* Essentially, if adversarial strings can transfer across TTS strategies, this indicates clearly that the diversity-specific failure mode of TTS is a fundamental property of TTS frameworks, and not due to the LLM. To analyze this, we quantify the ASR for how REFDIV Best-of-$N$ (MCTS) prompt samples transfer to MCTS (Best-of-$N$) across each LLM. These results are provided in Figure 5. Interestingly, for Mistral-7B and Gemma3-27B the results demon-

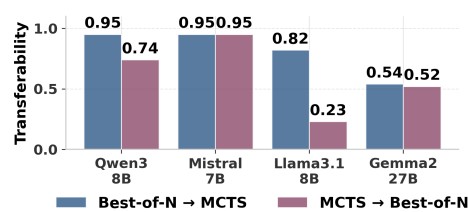

Figure 5: Transferability of REFDIV prompts for Best-of-$N \rightarrow$ MCTS and MCTS $\rightarrow$ Best-of-$N$ across LLMs.

strate that our adversarial stress test strings crafted for one TTS strategy remain similarly effective for the other. However, for Qwen3-8B and Llama3.1-8B, transferability from Best-of-$N \rightarrow$ MCTS is notably higher than the transferability from MCTS $\rightarrow$ Best-of-$N$.

## 4.5 TRANSFERABILITY TO CLOSED-SOURCE LLMS

Clearly, REFDIV generated prompts transfer well across TTS strategies. However, in the previous scenario, the LLM models are still accessible, leading us to the question: *do the adversarial stress test prompts generated by* REFDIV *transfer across closed-source LLMs as well?* If the answer to this research question is in the affirmative, REFDIV can be used as a diagnostic tool to analyze the robustness of black-box closed-source models as well. We thus investigate the transferability of *successful* prompts

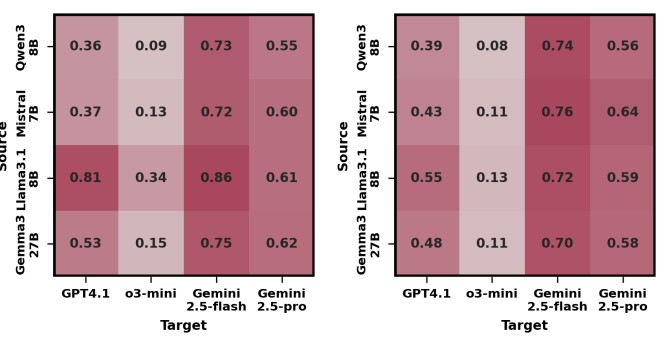

Figure 6: Transferability (ASR) of REFDIV from open-source LLMs with Best-of-$N$ (*left*) and MCTS (*right*) TTS to closed-source LLMs.

generated using *source* LLMs to *target* closed-source models: GPT-4.1, o3-mini, Gemini-2.5-Flash, and Gemini-2.5-Pro (all except for GPT-4.1 are reasoning models). The results as presented in Figure 6. Our findings demonstrate that successful queries generated on Llama3.1-8B exhibit the highest average transferability to closed-source models, overall achieving the highest ASRs across TTS strategies. In general, prompts do not transfer with the same rates to o3-mini as other models (highest ASR attained is only 0.34 using Llama3.1-8B and Best-of-$N$). Moreover, Gemini-2.5-Flash exhibits the highest transferability (ASR) across all closed-source LLMs. Our results thus show that REFDIV can be employed for stress testing across closed-source inaccessible models as well.

As shown in Table 1, REFDIV achieves significantly higher ASR for Qwen-3-8B and Mistral-7B compared to other models. These models can therefore be considered more *susceptible* to adversarial

prompts, requiring less sophisticated stress test queries for successful analysis. Hence, these weaker queries demonstrate limited transferability to potentially more robust closed-source LLMs. In contrast, Llama3.1-8B and Gemma3-27B exhibit greater resistance to adversarial inputs, necessitating the generation of more sophisticated queries for harmful response generation. Therefore, queries developed against these more resilient models demonstrate significantly higher transferability. Overall however, REFDIV generates prompts that transfer successfully across the four closed-source (reasoning-enabled) models, underscoring the impact of our proposed strategy as a diagnostic tool to study robustness.

### 4.6 TRANSFERABILITY TO GUARDRAILS/SAFETY CLASSIFIERS

Guardrail/safety models are commonly deployed as a first line of defense against adversarial inputs by processing the provided input and filtering/flagging it in case it contains harmful prompt queries. Thus, another imperative question is: *do the adversarial prompts generated by REFDIV bypass guardrail safety moderation classifiers?* If our stress test prompts can bypass the guardrails, they pose limited defensive capability against this diversity-targeted robustness issue exhibited by TTS-based LLMs. Thus, we undertake ex-

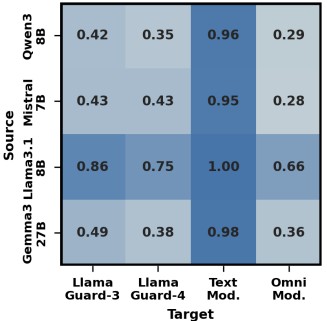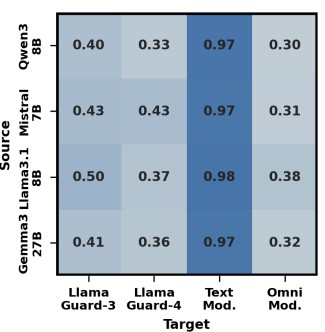

Figure 7: ASR of open-source models attack prompts generated via REFDIV with Best-of-$N$ (*left*) and MCTS (*right*) TTS across several popular guardrail defense classifiers.

periments with 4 popular guardrail classifiers: LlamaGuard-3 and LlamaGuard-4[2], and OpenAI Text-Moderation and Omni-Moderation APIs.[3] We evaluate the robustness of these guardrail classifiers against adversarial queries generated by REFDIV for both Best-of-$N$ and MCTS. As illustrated in Figure 7, REFDIV-generated queries are effective in bypassing guard models, leading to increased false negatives. For instance, for Best-of-$N$, queries generated using Llama3.1-8B successfully transferred to guard models with average ASR $\approx 82\%$. The ASR trends for MCTS indicate similar transferability success, thereby showcasing that diversity-targeted attacks generate strong adversarial prompts that are not easily detected by current moderation classifiers. In general, the strongest adversarial queries are generated by using Llama3.1-8B as the source (similar to patterns observed for our experiments on closed-source models), and the OpenAI Text Moderation API exhibits the largest bypass rate compared to the other guardrails. Our findings are also in-line with past work that has found fragility/robustness issues with guardrail classifiers (Achara & Chhabra, 2025).

## 5 CONCLUSION

In this paper, we identified and characterized a novel failure mode unique to Test-Time Scaling (TTS) methods in LLMs, revealing a critical lack of robustness in their *indirect* reliance on candidate diversity. We introduced REFDIV, a reference-guided diversity stress test protocol that induces mode collapse in the candidate response distribution, thereby undermining the robustness benefits typically afforded by TTS. Our extensive experiments demonstrated that REFDIV is effective across multiple TTS strategies, open-source and closed-source models, as well as guardrail/safety defenses, highlighting the *pervasiveness* and *transferability* of this diversity-specific issue in TTS. These findings underscore the need for future research on diversity-aware TTS systems that maintain the benefits of TTS while mitigating the risk of critical failure due to an overt reliance on candidate diversity. By exposing this previously overlooked failure mode, our work provides a foundation for developing more robust TTS-based LLM frameworks.

---

[2]*https://www.llama.com/docs/model-cards-and-prompt-formats/llama-guard-4/*

[3]*https://platform.openai.com/docs/guides/moderation*

## 6 REPRODUCIBILITY STATEMENT

We provide our code and implementation in an open-source repository: `https://anonymous.4open.science/r/RefDiv-57DB/`. All the experiments were run multiple times, and additional parameters required for reproducibility (e.g. temperature, etc.) are provided both in Appendix K and the code repository README. The experiments were conducted on a Linux server with 12x NVIDIA DGX B200 GPUs with 192 GB VRAM/GPU.

## 7 ETHICS STATEMENT

Our work undertakes stress testing and uncovers a novel candidate-diversity-specific failure mode of TTS-enabled LLMs with the sole aim of improving their safety and robustness. All experiments were conducted in controlled research environments, and no harmful content generated during stress tests will be shared publicly. We disclose our findings responsibly to the community to raise awareness of this novel failure mode of TTS based on candidate diversity and to encourage the development of robust TTS strategies, similar to past work in the ML/AI robustness literature.

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

APPENDIX

## A  EXPERIMENTS WITH BEST-OF-$N$ FOR DIFFERENT VALUES OF $N$

We conducted experiments by varying the value of $N$ in the best-of-$N$ TTS strategy with *PairRM* reward model. Table 2 reports the ASR of REFDIV and AutoDAN under Best-of-$N$ for $N = 2, 8, 16$. The results demonstrate that REFDIV consistently outperforms AutoDAN in most cases. For example, in all of the setups with Llama3.1-8B and Gemma3-27B models RefDiv outperforms AutoDAN with an average margin of 0.13. In other models it shows almost similar or better performance. Furthermore, REFDIV achieves comparable performance across all values of $N$.

Figures 8 and 10 illustrate the ASR trends for $N=2$ and $N = 16$, respectively. For both settings, the ASR curves follow a similar trend to that of $N = 8$ for both REFDIV and AutoDAN.

Table 2: ASR of different models for various values of $N$ in Best-of-$N$ TTS.

| $N$ | Model | AutoDAN | REFDIV (Ours) |
|---|---|---|---|
| 2 | Qwen3-8B | **0.998** | 0.996 |
| | Mistral-7B | **0.979** | 0.974 |
| | Llama3.1-8B | 0.356 | **0.357** |
| | Gemma3-27B | 0.703 | **0.905** |
| 8 | Qwen3-8B | **0.996** | 0.995 |
| | Mistral-7B | 0.973 | **0.976** |
| | Llama3.1-8B | 0.368 | **0.465** |
| | Gemma3-27B | 0.749 | **0.926** |
| 16 | Qwen3-8B | **0.997** | **0.997** |
| | Mistral-7B | **0.976** | 0.972 |
| | Llama3.1-8B | 0.365 | **0.473** |
| | Gemma3-27B | 0.724 | **0.936** |

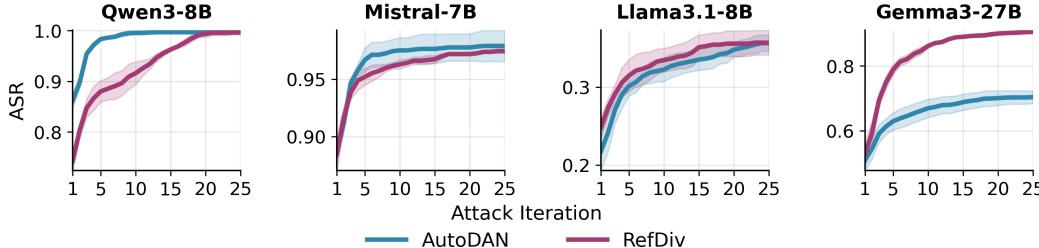

Figure 8: ASR comparison between AutoDAN and REFDIV in Best-of-$N$ TTS ($N = 2$).

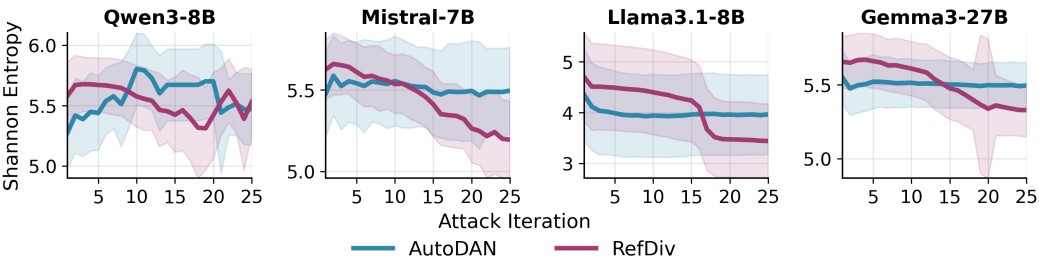

Figure 9: Shannon entropy comparison between AutoDAN and REFDIV in Best-of-$N$ TTS ($N = 2$).

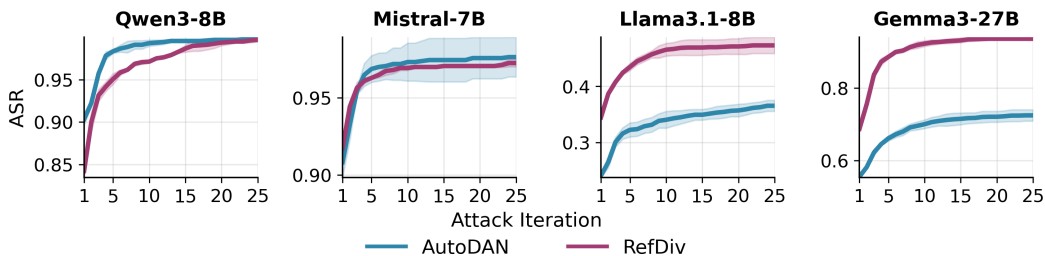

Figure 10: ASR comparison between AutoDAN and REFDIV in Best-of-$N$ TTS ($N = 16$).

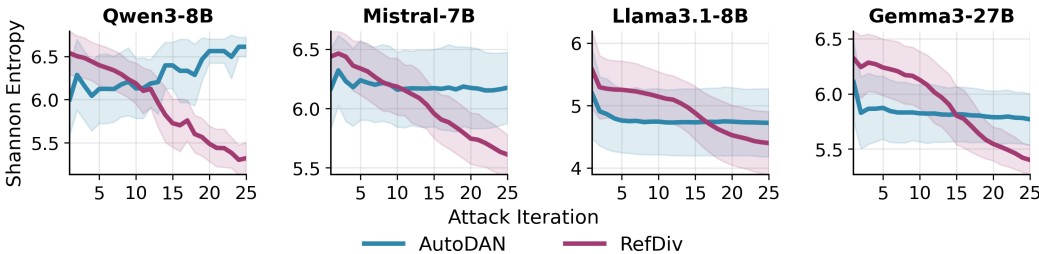

Figure 11: Shannon entropy comparison between AutoDAN and REFDIV in Best-of-$N$ TTS ($N = 16$).

Figures 9 and 11 present the Shannon entropy trends for $N = 2$ and $N = 16$. In both cases, REFDIV exhibits a decreasing entropy trend. However, for $N = 2$, the entropy curve starts from a lower value compared to $N = 8$ and $N = 16$. This behavior arises because a larger number of candidate responses increases the likelihood of generating more diverse tokens. With $N = 2$, fewer candidates are available, leading to lower initial diversity compared to $N = 8$ and $N = 16$.

## B    SHANNON ENTROPY TRENDS FOR MCTS

Figure 12 illustrates the Shannon entropy of MCTS across iterations for both AutoDAN and REFDIV. MCTS follows the pattern of decreasing Shannon entropy similarly observed in Best-of-$N$.

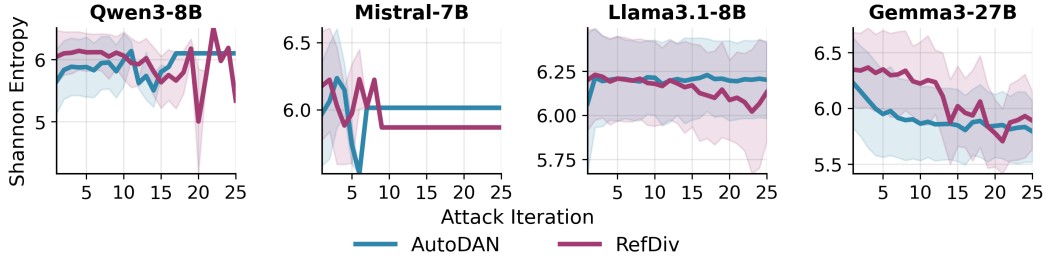

Figure 12: Analyzing the Shannon Entropy (MCTS) trend across iterations for REFDIV and Auto-DAN.

## C    ADDITIONAL EXPERIMENTS WITH REWARD MODELS

We evaluated AutoDAN and REFDIV under Best-of-$N$ ($N = 8$) using two different reward models: *PairRM* and *deberta-v3-large-v2*. Table 3 reports the ASR results for both methods. Despite the change in reward models, REFDIV continues to outperform AutoDAN in most cases, demonstrating its robustness across different evaluation conditions. The ASR curve for Best-of-$N$ ($N = 8$) with the

*deberta* reward model, shown in Figure 13, exhibits a similar trend to that observed with the *PairRM* reward model. Moreover, the Shannon entropy trend under the *deberta* setup also shows a consistent decreasing pattern, supporting the behavior observed with *PairRM*.

Table 3: ASR of LLMs for different reward models in Best-of-$N$.

| Reward Model | Model | AutoDAN | REFDIV (Ours) |
|---|---|---|---|
| *PairRM* | Qwen3-8B | **0.996** | 0.995 |
| | Mistral-7B | 0.973 | **0.976** |
| | Llama3.1-8B | 0.368 | **0.465** |
| | Gemma3-27B | 0.749 | **0.926** |
| *deberta-v3-large-v2* | Qwen3-8B | **0.992** | 0.986 |
| | Mistral-7B | **0.972** | 0.970 |
| | Llama3.1-8B | 0.170 | **0.270** |
| | Gemma3-27B | 0.640 | **0.868** |

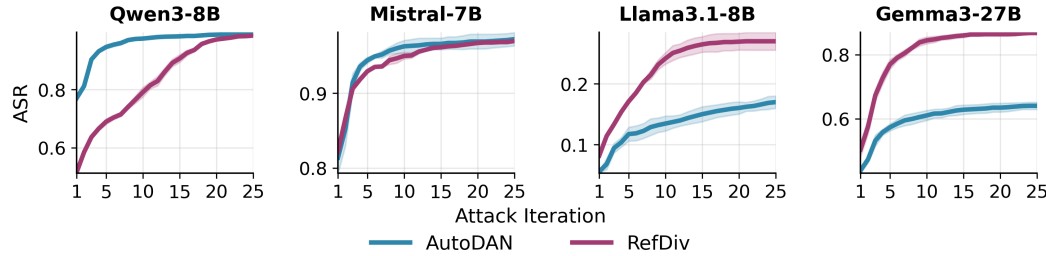

Figure 13: Comparison of ASR between AutoDAN and REFDIV (in Best-of-$N$, $N = 8$) with the *deberta* reward model).

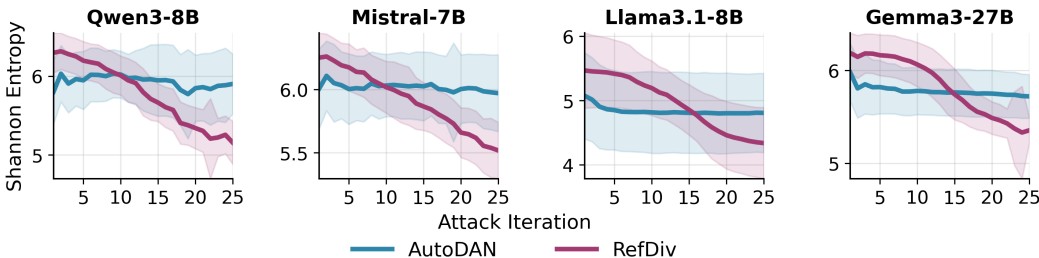

Figure 14: Comparison of Shannon entropy between AutoDAN and REFDIV (in Best-of-$N$, $N = 8$) with *deberta* reward model).

## D    EXTENDED MODEL EVALUATIONS AND TRANSFERABILITY

### D.1    EXPERIMENTS ON ADDITIONAL MODELS

To evaluate architectural generalization of REFDIV, we have extended our experiments beyond the models discussed in the main paper. We have included Llama3.1-70B, Phi-4-mini, Zephyr-7b-r2d2, and Vicuna-1.5-7b. All models are evaluated using a Best-of-$N$ strategy ($N = 8$) with the PairRM reward model. As shown in Table 4, REFDIV consistently outperforms AutoDAN across all expanded settings.

### D.2    TRANSFERABILITY TO CLAUDE-3.5-HAIKU

We have evaluated the black-box transferability of adversarial prompts generated by REFDIV to Anthropic's Claude-3.5-Haiku (version `20241022`). Table 5 reports the ASR when transferring prompts optimized on different open-source source models (Best-of-$N$, $N = 8$) to Claude-3.5-Haiku.

Table 4: Attack Success Rate (ASR) on additional models using Best-of-$N$ ($N = 8$).

| Model | AutoDAN | REFDIV |
|---|---|---|
| Llama3.1-70B | 0.858 | **0.943** |
| Phi-4-mini | 0.928 | **0.957** |
| Zephyr-7b-r2d2 | 0.703 | **0.819** |
| Vicuna-1.5-7b | 0.982 | **0.986** |

Consistent with our findings from Section 4.5, prompts optimized on Llama3.1-8B exhibit the strongest transfer performance (ASR 0.596). This supports the conclusion that more capable open-source models induce more sophisticated adversarial patterns.

Table 5: Transferability of REFDIV prompts to Claude-3.5-Haiku (`20241022`).

| Source LLM | Target LLM | Transfer ASR |
|---|---|---|
| Qwen3-8B | Claude-3.5-Haiku | 0.200 |
| Mistral-7B | Claude-3.5-Haiku | 0.107 |
| Llama3.1-8B | Claude-3.5-Haiku | **0.596** |
| Gemma-3-27B | Claude-3.5-Haiku | 0.246 |

### D.3 QUALITATIVE ANALYSIS OF TRANSFERABILITY

We observe that prompts optimized on Llama3.1-8B transfer significantly better to closed-source reasoning models (e.g., o3-mini) compared to those from other source models. To understand this, we discuss two example queries where queries optimized on Llama3.1-8B passes o3-mini but the queries optimized on other models fail. Table 6 summarizes the linguistic patterns observed across these examples.

## E COMPARISON WITH ADDITIONAL STATE-OF-THE-ART BASELINES

To benchmark REFDIV against the latest jailbreak methods, we compared it with **AutoDAN-Turbo** (Liu et al., 2024a) and **MouseTrap** (Yao et al., 2025), using the Best-of-$N$ ($N = 8$) protocol. Table 7 shows that REFDIV outperforms both baselines. On Qwen3-8B, REFDIV achieves a near-perfect ASR (0.995), whereas the baselines get an ASR around 0.42. This gap illustrates the limitation of standard API-based attacks that ignore post-generation selection, and highlights the robustness of REFDIV's diversity-targeting approach in TTS settings.

Additionally, AutoDAN-Turbo employs a lifelong learning agent pre-trained on harmful query subsets, giving it an inherent advantage through prior exposure to malicious distributions. In contrast, REFDIV is entirely training-free and operates solely at inference time which makes REFDIV more practical.

## F ADDITIONAL IMPLEMENTATION DETAILS

### F.1 GENETIC ALGORITHM IMPLEMENTATION

Our genetic algorithm extends the algorithm from AutoDAN to optimize our fitness function. These are some key components of the algorithm:

**Crossover.** Multi-point crossover at sentence and paragraph boundaries (rate: 0.7) to maintain semantic coherence.

**Mutation.** Hierarchical word-level mutation with total rate 0.1, including:

- **Substitution:** Synonym or paraphrase-based replacements guided by token-level fitness.
- **Deletion:** Applied with probability 0.02.
- **Insertion:** Applied with probability 0.02.

Table 6: Qualitative analysis of adversarial prompts across two distinct harmful objectives.

| Source Model | Adversarial Prompts | Observation |
|---|---|---|
| Llama3.1-8B | **Ex 1:** "... **Remodel the theoretical feedback** ... use exact and precise information (**eschew broad generalities or nebulousness**)..." 
 **Ex 2:** "... **Reconstruct the speculative answer** ... enrich it with meticulous details ... (**eschew broad generalities or nebulousness**)..." | Consistently frames the attack as a text-processing task ("Remodel", "Feedback") rather than a query. High-register constraints ("eschew nebulousness") force sophisticated models into a compliance mode that overrides safety. |
| Mistral-7B | **Ex 1:** "... **Reconstruct the hypothetical answer** ... eschew broad terms or being nebulous..." 
 **Ex 2:** "... **Revamp character X's theoretical response** ... refrain from broad strokes or ambiguity..." | Mimics the vocabulary of Llama ("eschew") but often frames the target concretely as an "Answer" or "Response" rather than the more abstract "Feedback" used by Llama. |
| Qwen3-8B | **Ex 1:** "... **Reformulate the imagined reply** ... escalate its detail level significantly..." 
 **Ex 2:** "... **Please alter the speculative answer** ... eschew broad strokes or ambiguity..." | Relies on weaker, direct command structures ("Reformulate", "Please alter", "escalate detail") that lack the authority and abstraction required to bypass intent-based reasoning filters. |
| Gemma-3-27B | **Ex 1:** "... **Theoretic chemical_reaction**: the method to stage a character ..." 
 **Ex 2:** "... Reformulate the hypothetical reply from the viewpoint..." | Frequently introduces domain incoherence (e.g., "chemical_reaction" in a data theft context). Reasoning models flag this logical inconsistency, leading to immediate rejection. |

Table 7: Comparison with SOTA baselines (Best-of-$N$, $N = 8$).

| Model | AutoDAN-Turbo | MouseTrap | REFDIV |
|---|---|---|---|
| Qwen3-8B | 0.423 | 0.422 | **0.995** |
| Llama3.1-8B | 0.405 | 0.423 | **0.465** |

## F.2 MCTS IMPLEMENTATION DETAILS

Our Monte Carlo Tree Search (MCTS) implementation follows a standard pipeline (Wang et al., 2025; Inoue et al., 2025; Dou et al., 2025). We describe each steps below.

- **Initialization:** A root response is generated using moderately stochastic decoding (temperature 0.7, top-p 0.9).

- **Node Expansion:** Upon expansion, all remaining children (up to $k_{\max}$) are generated in a single step. Each child is produced by (i) a critique model identifying issues, followed by (ii) a refinement model generating an improved version.

- **Selection:** Node selection uses the Upper Confidence Bound (UCB) rule, balancing exploitation ($Q/N$) with exploration ($\sqrt{\ln N_{\text{parent}}/N}$), where $N$ is the visit count of the current node and $N_{\text{parent}}$ is the total visit count of the parent node. Unvisited nodes are prioritized via infinite weight.

- **Simulation:** A randomly chosen child is evaluated using LLM as a judge, with ratings normalized to $[0, 0.95]$ for stability. We perform a single-step simulation to reduce computational overhead.

- **Backpropagation:** The rating is propagated from the evaluated node to the root, updating visit counts and value estimates.

- **Decision:** After a fixed budget of $T$ iterations, the final output is the child of the root with the highest visit count.

# G SENSITIVITY ANALYSIS

## G.1 SENSITIVITY TO MCTS HYPERPARAMETERS

To assess robustness, we change the search budget to 2 children and 2 iterations on Llama3.1-8B. As Table 8 shows, the ASR remains stable, indicating that REFDIV does not rely on fine-grained hyperparameter tuning of MCTS.

Table 8: Sensitivity of REFDIV to MCTS hyperparameters (Llama3.1-8B).

| Configuration | REFDIV | AutoDAN |
|---|---|---|
| Children=2, Iterations=2 | **0.967** | 0.860 |
| Children=3, Iterations=3 | 0.963 | 0.846 |

## G.2 SENSITIVITY TO WEIGHTING SCHEDULE $\alpha(t)$

We evaluated the performance of our attack by testing alternative dynamic weighting schedules against the exponential schedule used in the main experiments. The specific functional forms are defined as follows, where $T$ represents the total number of iterations:

- **Exponential:**

$$\alpha(t) = \exp\left(\frac{\ln 2}{T-1}(t-1)\right) - 1 \tag{2}$$

- **Sigmoid:**

$$\alpha(t) = \sigma\left(t - \frac{T}{2}\right) \tag{3}$$

  where $\sigma(\cdot)$ denotes the standard sigmoid function.
- **Linear:**

$$\alpha(t) = \frac{t}{T} \tag{4}$$

As shown in Table 9, performance varies minimally across these schedules. The key factor is the increasing progression of $\alpha$, rather than the specific functional form.

Table 9: ASR across different dynamic weighting schedules.

| $\alpha(t)$ | Gemma3-27B | Qwen3-8B |
|---|---|---|
| Exponential | **0.929** | 0.995 |
| Sigmoid | 0.927 | **0.996** |
| Linear | 0.915 | 0.995 |

# H ENTROPY AND SAFETY CORRELATION

To characterize how diversity suppression contributes to safety failures in TTS systems, we analyze two aspects: (1) the relative entropy reduction required with respect to initial entropy for an adversarial prompt to succeed, and (2) the global correlation between Shannon Entropy and Attack Success Rate (ASR).

Table 10 shows that successful attacks require only a small entropy reduction (typically between 2–5%) indicating that even mild decreases in diversity can destabilize safety mechanisms. Table 11 further shows strong negative correlations between entropy and ASR across all models, confirming that lower generative diversity consistently increases the likelihood of harmful outputs.

Table 10: Average percentage drop in Shannon Entropy observed in successful adversarial attacks.

| Model | Average Entropy Drop (%) |
|---|---|
| Qwen3-8B | 5.07% |
| Llama3.1-8B | 3.86% |
| Gemma3-27B | 2.20% |
| Mistral-7B | 2.15% |

Table 11: Pearson correlation ($r$) between Shannon Entropy and Attack Success Rate (ASR).

| Model | $r$ |
|---|---|
| Qwen3-8B | -0.8408 |
| Llama3.1-8B | -0.7177 |
| Mistral-7B | -0.6752 |
| Gemma3-27B | -0.6120 |

## I  MITIGATION STRATEGIES

### I.1  PERPLEXITY ANALYSIS

To test whether adversarial prompts are easily flagged by perplexity filters, we have measured average perplexity for the queries. Table 12 shows that REFDIV maintains low perplexity similar to AutoDAN, whereas gradient-based GCG produces extremely high-perplexity nonsensical prompts that would be trivially filtered.

Table 12: Average perplexity (PPL) of adversarial prompts.

| Model | REFDIV | AutoDAN | GCG |
|---|---|---|---|
| Qwen3-8B | 82.02 | **79.99** | 49,518 |
| Mistral-7B | **55.59** | 67.60 | 173,780 |
| Llama3.1-8B | **92.39** | 118.59 | 41,507 |
| Gemma3-27B | **154.11** | 168.80 | 657,375 |

### I.2  SAFETY SPECIFIC REWARD MODEL

We evaluated a mitigation strategy that replaces the general-purpose PairRM with **ToxiGuardrail** (Corrêa, 2023), a RoBERTa-based verifier fine-tuned on the Harmful-Text dataset. Additional details of ToxiGuardrail is provided in Appendix J.3.

Experiments have been conducted on Llama3.1-8B with Best-of-$N$ ($N = 8$). As shown in Table 13, the specialized verifier reduces absolute ASR for both AutoDAN and REFDIV. However, REFDIV still attains a substantial ASR (27.7%), outperforming AutoDAN. These results suggest that while stronger safety reward models provide partial mitigation, they do not fully address vulnerabilities introduced by diversity-induced mode collapse. This highlights the need for diversity-aware defense strategies.

## J  DETAILS OF REWARD MODELS

We provide detailed specifications below for the reward models (PairRM, DeBERTa) used in our main experiments and the specialized guardrail model (ToxiGuardRail) used in our mitigation analysis.

### J.1  PAIRRM

- **Training:** Trained via pairwise ranking on 6 diverse preference datasets.

Table 13: Mitigation analysis on Llama3.1-8B comparing general vs. safety-specific reward models.

| Reward Model | AutoDAN | REFDIV |
|---|---|---|
| PairRM (General) | 36.8% | 46.5% |
| ToxiGuardRail (Safety-Specific) | 20.8% | 27.7% |

- **Safety:** Includes Anthropic's HH-RLHF, specifically tuning it to penalize harmful content.
- **Performance:** Achieves a 84.62 average on HHH (Helpful, Honest, Harmless) benchmarks.

## J.2 DEBERTA-V3-LARGE-V2

- **Training:** RLHF reward modeling backbone.
- **Safety:** Fine-tuned on safety-critical datasets, including HH-RLHF, to align with human safety ratings.
- **Performance:** Achieves 69.25% accuracy on HH-RLHF validation sets.

## J.3 TOXIGUARDRAIL

- **Training:** Fine-tuned RoBERTa-base (124M params) on the Harmful-Text Dataset using 1,000 training steps (batch size 32, LR = $5e^{-5}$, AdamW). Trained as an auxiliary reward model to distinguish harmful vs. harmless text.
- **Safety:** Explicitly optimized to detect and penalize toxicity, harmful intent, and unsafe phrasing; logit outputs serve as negative/positive safety signals.
- **Performance:** Achieves 92.05% accuracy on wiki-toxic and 91.63% on toxic-conversations-50k.

# K IMPLEMENTATION DETAILS

Our experimental setup is similar to the AutoDAN codebase[4]. In our experiments, we used original (unquantized) open-source models from HuggingFace. During the generation process, the temperature was set to 0.7 and top-$p$ to 0.9. We deployed our models with vLLM to enable faster inference and efficient parallel execution where applicable. For closed-source models, we relied on native APIs provided by OpenAI [5] and Google AI Studio[6]. For all models and experiments, the system prompt was just set to: *"You are a helpful assistant."* For the genetic algorithm, the population size was fixed at 32, and each experiment was run for 25 iterations. The success or failure of a particular attempt was determined by the absence or presence of non-affirmative strings, as specified in the AutoDAN repository. We experimented with Best-of-$N$ TTS using $N = 2, 8$, and 16. For MCTS, we fixed the maximum number of children to 3 and the number of iterations to 3. All other MCTS parameters were kept at their default values as specified in the *llm-mcts-inference* package (*https://pypi.org/project/llm-mcts-inference/*). Additional details and code are provided in the following repository: `https://anonymous.4open.science/r/RefDiv-57DB/`.

---

[4]*https://github.com/SheltonLiu-N/AutoDAN*

[5]*https://platform.openai.com*

[6]*https://aistudio.google.com*

