# OpenReview forum: "Less Diverse, Less Safe: The Indirect But Pervasive Risk of Test-Time Scaling in Large Language Models"
_ICLR.cc/2026/Conference — Submitted to ICLR 2026_

### Official Review · Reviewer_zwHe · 2025-10-26

**Soundness:** 2
**Presentation:** 3
**Contribution:** 2
**Rating:** 4
**Confidence:** 4

**Summary:**

This paper identifies a novel failure mode in Test-Time Scaling (TTS) methods for LLMs: when candidate diversity is reduced, TTS becomes significantly more vulnerable to producing unsafe outputs. The authors propose REFDIV (Reference-Guided Diversity Stress Test Protocol), a genetic algorithm-based attack that constrains candidate diversity using Shannon entropy minimization. Through extensive experiments across multiple open-source models (Qwen3, Mistral, Llama3.1, Gemma3), TTS strategies (Best-of-N, MCTS), and closed-source systems (o3, Gemini-2.5), they demonstrate that diversity reduction consistently increases attack success rates, often surpassing existing jailbreak methods. The attacks also successfully bypass popular safety guardrails like Llama-Guard and OpenAI Moderation API.

**Strengths:**

1. The paper identifies a new vulnerability specific to TTS methods. The connection between candidate diversity and safety is intuitive yet previously unexplored, making this a valuable contribution as TTS becomes more prevalent in production systems.
2. REFDIV consistently outperforms baselines, particularly on harder models (Llama3.1-8B: 0.465 vs 0.368; Gemma3-27B: 0.926 vs 0.749 for Best-of-N), and shows reasonable transferability to closed-source systems.

**Weaknesses:**

1. REFDIV is essentially AutoDAN with a diversity-focused fitness function. The genetic algorithm framework is standard; the main contribution is recognizing diversity as a target rather than methodological innovation. The dynamic weighting scheme α(t) appears somewhat ad-hoc without justification for the specific exponential schedule.
2. The paper identifies the vulnerability but offers no mitigation strategies. Suggestions for diversity-preserving TTS variants, entropy-based monitoring, or robustness enhancements would strengthen the contribution.
3. Recent reasoning-model attacks mentioned in related work (Mousetrap, AutoRAN, H-CoT) are not evaluated. Without these comparisons, it's unclear whether REFDIV represents state-of-the-art or merely improves on AutoDAN.

**Questions:**

1. Have you explored diversity-aware defenses? For example, could TTS explicitly maintain entropy thresholds, or could reward models penalize low-diversity candidate sets?
2. Why does REFDIV show much larger improvements over AutoDAN for Llama3.1-8B and Gemma3-27B? Are certain alignment techniques more susceptible to diversity attacks?
3. Are REFDIV-generated prompts more or less natural/detectable than AutoDAN prompts? Could simple perplexity filters help?

---

> ### Author Response · Authors · 2025-11-22
> **Reviewer zwHe - 1/3**
>
> Dear reviewer zwHe,\
> Thank you for putting your time and effort into reviewing our paper. We are providing clarifications to the comments raised below.
> ___
> ***W-1: REFDIV is essentially AutoDAN with a diversity-focused fitness function. The genetic algorithm framework is standard; the main contribution is recognizing diversity as a target rather than methodological innovation. The dynamic weighting scheme α(t) appears somewhat ad-hoc without justification for the specific exponential schedule.***
> ___
> **WR-1:** We agree that the genetic algorithm framework is adapted from AutoDAN; however, our contribution lies in the **objective function**, not the optimizer. We purposely utilized a standard optimizer to isolate and demonstrate that **targeting candidate diversity** (via our novel fitness function) is the specific mechanism that breaks TTS defenses.
> Regarding the dynamic weighting scheme α(t), we provide the following justifications:
> - **Curriculum Learning Strategy:** The schedule is not ad-hoc; it implements a specific curriculum. It transitions from **Reference-Guided Exploration** (high guidance) to **Diversity-Minimization Exploitation** (pure mode collapse), ensuring the attack finds the adversarial subspace before converging.
> - **Empirical Robustness:** To prove the schedule is robust rather than arbitrary, we compared our **Exponential** schedule against **Sigmoid** and **Linear** variants. As shown below, performance is stable across all increasing functions.
>
> | **$\alpha(t)$ Method** | **Gemma3-27B** | **Qwen3-8B** |
> | :--- | :--- | :--- |
> | Exponential | **0.929** | 0.995 |
> | Sigmoid | 0.927 | **0.996** |
> | Linear | 0.915 | 0.995 |
>
> **Functions:**
> - $$\textbf{Exponential: } \alpha(t) = \exp\left( \frac{\ln(2)}{T-1} \cdot (t-1) \right) - 1$$
>
> - $$ \textbf{Sigmoid: } \alpha(t) = \text{sigmoid}\left( t - \frac{t_{\max}}{2} \right) $$
>
> - $$ \textbf{Linear: } \alpha(t) = \frac{t}{t_{\max}} $$
>
> ___
> ___
>
> ***W-2: The paper identifies the vulnerability but offers no mitigation strategies. Suggestions for diversity-preserving TTS variants, entropy-based monitoring, or robustness enhancements would strengthen the contribution.***
> ___
> **WR-2:** We appreciate the suggestion to explore mitigation strategies. To this end, we evaluated the effectiveness of **Safety-Specific Reward Models** as a defense mechanism, specifically deploying **ToxiGuardrail** (a toxicity-focused scorer) in place of the general-purpose reward models like PairRM. We provide the details of the model and experiment below.
> - **ToxiGuardRail (Kluge Corrêa, 2023):**
>     - **Training:** Fine-tuned **RoBERTa-base** (124M params) on the **Harmful-Text Dataset** using 1,000 training steps (batch size 32, LR = 5e-5, AdamW). Trained as an auxiliary reward model to distinguish harmful vs. harmless text.
>     - **Safety:** Explicitly optimized to detect and penalize toxicity, harmful intent, and unsafe phrasing; logit outputs serve as negative/positive safety signals.
>     - **Performance:** Achieves **92.05%** accuracy on wiki_toxic and **91.63%** on toxic_conversations_50k.
> - **Mitigation Experiment:** We tested this defense on **Llama-3.1-8B** using the Best-of-$N$ ($N$=8) strategy.
>
> | Guardrail | AutoDAN | RefDiv |
> | :--- | :--- | :--- |
> | ToxiGuardRail | 20.8% | 27.7% |
> | PairRM | 36.8% | 46.5% |
>
>
> - **Effective Mitigation:** Deploying a specialized safety reward model significantly reduces the absolute Attack Success Rate (dropping RefDiv from 46.5% to 27.7%).
> - **Partial Solution:** While effective, it does not fully resolve the vulnerability. RefDiv still outperforms AutoDAN, indicating that even with robust safety scoring, the underlying **diversity-constrained mode collapse** issue forces the TTS selector to accept harmful outputs when diversity is sufficiently constrained.
>
> ___

---

> > ### Author Response · Authors · 2025-11-22
> > **Reviewer zwHe - 2/3**
> >
> > ___
> >
> > ***W-3: Recent reasoning-model attacks mentioned in related work (Mousetrap, AutoRAN, H-CoT) are not evaluated. Without these comparisons, it's unclear whether REFDIV represents state-of-the-art or merely improves on AutoDAN.***
> >
> > ___
> >
> > **WR-3:** Thank you for suggesting these additional state-of-the-art baselines. We acknowledge that the field moves rapidly, and comparing against recent methods like **MouseTrap** [1] strengthens our evaluation. Furthermore, we extended our evaluation to also include **AutoDAN-Turbo** [2].
> >
> > - **MouseTrap Comparison:** Our results indicate that RefDiv achieves higher attack success rates compared to MouseTrap. This validates that while MouseTrap effectively optimizes individual prompts for harmfulness, it does not account for the **candidate diversity** required to bypass the TTS verification layer. We utilized the official implementation with Llama-3-8B acting as the Judge-LLM.
> >
> > - **AutoDAN-Turbo Comparison:** We have also successfully reproduced AutoDAN-Turbo using the official codebase. It is important to note that AutoDAN-Turbo has an unfair advantage as it is **training-based** (training on ~50 harmful queries from HarmBench), whereas RefDiv is training-free and never sees this data. Despite this disadvantage, RefDiv constitutes a general-purpose TTS attack that constrains diversity and attains better ASR.
> >
> > **Results (Best-of-$N$, $N$=8):**
> >
> > | Model | MouseTrap | AutoDAN-Turbo | **RefDiv (Ours)** |
> > | :--- | :--- | :--- | :--- |
> > | **Qwen3-8B** | 42.21% | 41.28% | **99.50%** |
> > | **Llama3.1-8B** | 42.27% | 44.07% | **46.50%** |
> >
> > - **Analysis:**
> > RefDiv outperforms both baselines. This happens because MouseTrap and AutoDAN-Turbo optimize for generation probability in a single turn. They do not account for the post-generation selection mechanism of TTS. RefDiv specifically **targets the candidate pool distribution** (mode collapse), allowing it to bypass the TTS selection filter more effectively.
> >
> > **References:**\
> > [1] Yao et al., “A Mousetrap: Fooling Large Reasoning Models for Jailbreak with Chain of Iterative Chaos,” Findings of ACL 2025.\
> > [2] Liu et al., “AutoDAN-Turbo: A Lifelong Agent for Strategy Self-Exploration to Jailbreak LLMs,” ICLR 2025 (Spotlight).
> >
> >
> > ___
> > ___
> >
> > ***Q-1: Have you explored diversity-aware defenses? For example, could TTS explicitly maintain entropy thresholds, or could reward models penalize low-diversity candidate sets?***
> > ___
> > **QR-1:** Thank you for this insightful suggestion regarding diversity-aware defenses. We provide some justifications regarding this point below:
> > - **Currently Implemented Diversity Constraints (Varying $N$):** The number of candidate samples ($N$) in Best-of-$N$ or MCTS is a trivial mechanism to control diversity (i.e., fewer candidates implies less likelihood of token diversity). However, our results in **Table-2** of Appendix A demonstrate that varying $N$ (from 2 to 16) does not mitigate the attack, as REFDIV remains consistently effective. **Figure-2,4,8-11** show similar trends for both ASR and Shannon Entropy across iterations for varying $N$. This indicates that simple diversity defenses are unlikely to succeed.
> > - **Task Performance Trade-offs:** Enforcing strict diversity or entropy thresholds at the token-level is non-trivial because it risks arbitrarily worsening performance on tasks that inherently require high generation diversity, such as creative writing.
> > - **Future Work:** Consequently, developing sophisticated diversity-based defenses is an important but complex challenge we aim to tackle in future work. The focus of this work was to propose REFDIV as a diagnostic stress test tool that exposes the hidden reliance of TTS on candidate diversity, demonstrating that when this diversity is compromised, even robust models fail. In accordance with the reviewer’s point, this will be our next aim and we will undertake it in a more sophisticated manner than our current approach with varying $N$.
> > ___

---

> > > ### Author Response · Authors · 2025-11-22
> > > **Reviewer zwHe - 3/3**
> > >
> > > ___
> > > ***Q-2: Why does REFDIV show much larger improvements over AutoDAN for Llama3.1-8B and Gemma3-27B? Are certain alignment techniques more susceptible to diversity attacks?***
> > > ___
> > > **QR-2:** The performance gap arises because Llama 3.1 and Gemma 3 employ intrinsic alignment techniques that reduce output entropy, creating a specific vulnerability to RefDiv's diversity constraints:
> > > - **Llama 3.1:** Enforces rigid safety via **Safety-Specific DPO** , **Rejection Sampling** , and **Refusal Tone Guidelines**. [2]
> > > - **Gemma 3:** Inherits strict safety distributions through **Knowledge Distillation** and **Safety Policy RLHF**. [1]
> > > Unlike **Mistral 7B** (which relies on inference-time **System Prompts** ) [3] or **Qwen3-8B** (which integrates safety via generalized training ) [4], these models are aligned to be "safe-by-design." This structural rigidity eliminates the diverse "refusal" candidates that standard attacks like AutoDAN rely on, allowing RefDiv to succeed by minimizing entropy and candidate diversity.
> > >
> > >
> > > **References:**\
> > > [1] [Gemma3 Technical Report](https://arxiv.org/html/2503.19786v1)\
> > > [2] [Grattafiori et al., The Llama 3 Herd of Models, 2024](https://arxiv.org/abs/2407.21783)\
> > > [3] [Jiang et al., Mistral 7B](https://arxiv.org/abs/2310.06825)\
> > > [4] [Yang et al., Qwen3 Technical report](https://arxiv.org/abs/2505.09388)
> > >
> > > ___
> > > ___
> > >
> > > ***Q-3: Are REFDIV-generated prompts more or less natural/detectable than AutoDAN prompts? Could simple perplexity filters help?***
> > > ___
> > > **QR-3:** Thanks for the great suggestion. RefDiv-generated prompts exhibit **naturalness comparable to AutoDAN**, making them difficult to detect with simple perplexity filters.
> > > - **Perplexity Analysis:** We measured the average perplexity (PPL) of adversarial queries across four models. As shown below, RefDiv maintains low perplexity scores (similar to AutoDAN), whereas gradient-based methods like GCG produce highly unnatural, easily detectable gibberish (PPL > 40k).
> > >
> > >
> > > | **Model** | **RefDiv** | **AutoDAN** | **GCG** |
> > > | :--- | :--- | :--- | :--- |
> > > | Qwen3-8B | 82.02 | **79.99** | 49,518.23 |
> > > | Mistral-7B | **55.59** | 67.60 | 173,780.06 |
> > > | Llama3.1-8B | **92.39** | 118.59 | 41,507.12 |
> > > | Gemma3-27B | **154.11** | 168.80 | 657,375.87 |
> > >
> > >
> > > - Since RefDiv prompts achieve low perplexity (even lower than AutoDAN on Llama3.1 and Gemma3), they are linguistically exceptionally coherent. In essence, using perplexity filters, RefDiv attack prompts are even harder to detect, much more compared to AutoDAN, which is already very sophisticated. Therefore, **simple perplexity thresholds would fail** to flag them as adversarial (unlike GCG queries).
> > >
> > > ___

---

### Official Review · Reviewer_vxQJ · 2025-10-31

**Soundness:** 3
**Presentation:** 2
**Contribution:** 2
**Rating:** 4
**Confidence:** 3

**Summary:**

This paper investigates a novel failure mode in Test-Time Scaling (TTS) methods for Large Language Models, focusing on how reduced candidate diversity can compromise safety. The authors propose REFDIV (Reference-Guided Diversity Stress Test Protocol), a reference-guided diversity stress-testing attack that constrains the diversity of candidate responses generated during test-time, thereby steering TTS-enabled LLMs toward producing unsafe outputs. The method uses a genetic algorithm with a diversity-guided fitness function that combines generation diversity (Shannon entropy) and reference-guided diversity (alignment with affirmative tokens). Experiments across open-source models on two TTS strategies (Best-of-N and MCTS) demonstrate that REFDIV achieves higher Attack Success Rates compared to baseline jailbreak methods like GCG and AutoDAN. The adversarial prompts also transfer across TTS strategies, to closed-source models and bypass popular safety guardrails (e.g., LlamaGuard-3/4).

**Strengths:**

+ Novel Failure Mode Identification: The paper identifies a previously unrecognized vulnerability in TTS methods—their implicit reliance on candidate diversity.

+ Comprehensive Experimental Validation: The authors conduct experiments across multiple dimensions: different TTS strategies (Best-of-N + reward models, MCTS), both open-source and closed-source models, and multiple safety guardrails. This thorough evaluation demonstrates the generality and pervasiveness of the identified failure mode.

+ Strong Transferability Results: REFDIV demonstrates strong transferability across TTS strategies, to black-box closed-source models, and across safety classifiers. This transferability suggests the diversity-driven vulnerability is a fundamental property of TTS frameworks rather than model-specific artifacts.

**Weaknesses:**

+ Unclear Threat Model for TTS: The paper does not clearly articulate the threat model specific to TTS systems. While the white-box assumption is stated (access to M and T but not explicit access to reward model r), it's unclear: (a) what adversarial capabilities are realistic in deployed TTS systems, (b) whether attackers would have access to the TTS strategy details in practice, and (c) how this threat model compares to standard LLM threat models. The motivation for why an adversary would specifically target diversity rather than directly optimizing for harmful outputs needs clearer justification.

+ Missing Technical Details and Result Analysis:

  1. Genetic Algorithm Details: The offspring generation process through "crossover and mutation" (Algorithm 1, line 10) is not specified. How are prompts crossed over? What mutation operators are used? These details are critical for reproducibility and should be in the main body of the paper.

  2. Reward Model Training and Safety: The paper uses PairRM and deberta-v3-large-v2 as reward models but provides no information about: (i) how these models were trained, (ii) whether they were specifically trained for safety, (iii) their baseline safety performance, and (iv) whether they might inadvertently contribute to the vulnerability.

  3. Entropy Analysis: While Figure 4 shows Shannon entropy decreases for REFDIV, the paper lacks deeper analysis of: (i) what specific entropy thresholds lead to safety failures, (ii) how entropy correlates with ASR quantitatively, and (iii) why some models (e.g., Llama3.1-8B) are more sensitive to diversity reduction than others.

  4. Transferability Mechanisms: The paper observes that prompts from Llama3.1-8B transfer better to closed-source models but only provides a high-level explanation. What specific properties of these prompts enable better transfer?


+ Baseline Comparisons Not State-of-the-Art: The paper compares against GCG and AutoDAN, but these are not the most recent state-of-the-art jailbreaking methods. Notable omissions include:
  1. PAIR (Chao et al., 2023) - which is cited but not used as a baseline despite being more effective than GCG
  2. AutoDAN-Turbo (Liu et al., 2024) - which is cited and explicitly described as extending AutoDAN with "lifelong learning," yet only the original AutoDAN is used for comparison

**Questions:**

+ MCTS Clarification: Can you provide a more detailed explanation of how MCTS is used as a TTS strategy in Section 4.1? Specifically: How does MCTS explore the solution space of LLM generations? The paper states "each instantiation is run with default parameters for the number of children (=3), for a total of 3 MCTS iterations" (Section 4.1). This requires justification: Why were these specific values chosen (3 children, 3 iterations)? How sensitive are the results to these hyperparameters?

+ Fitness Function Design: The dynamic weighting α(t) transitions from reference-guided to intrinsic diversity. What is the sensitivity to the specific exponential schedule? Have you tried other schedules (linear, sigmoid, etc.)?

+ Reward Model Vulnerability: To what extent does the vulnerability stem from the reward model itself versus the TTS strategy? Could adversarially robust reward models mitigate this attack?

+ Defense Mechanisms: Beyond identifying the vulnerability, what potential defenses could mitigate diversity-targeted attacks? For example, could diversity-aware safety filtering or ensemble-based verification help?

+ Real-world Impact: In practical deployments, how likely is it that adversaries would have the white-box access required for REFDIV? Could the method be adapted to black-box settings?

---

> ### Author Response · Authors · 2025-11-22
> **Reviewer vxQJ - 1/7**
>
> Dear reviewer vxQj,
>
> Thank you for sharing your observations on our paper. We really appreciate your effort to review our paper. We are providing some clarifications based on your observations.
> ___
> ***W-1: Unclear Threat Model for TTS: The paper does not clearly articulate the threat model specific to TTS systems. While the white-box assumption is stated (access to M and T but not explicit access to reward model r), it's unclear: \
> (a) what adversarial capabilities are realistic in deployed TTS systems, \
> (b) whether attackers would have access to the TTS strategy details in practice***
> ___
> **WR-1.a&b:** Thank you for this point, we will further emphasize our threat models and the aforementioned points in the revised version of the paper. To directly answer the questions above, we discuss the threat model based on two distinct operational scenarios:
> - **Scenario A: Diagnostic Stress Testing (White-Box)**
>     - This is the primary setting for **RefDiv as a tool** for developers and red-teams auditing their own systems.
>     - **Assumption:** The auditor has access to the Model (M) and the Scaling Strategy (T), allowing them to measure candidate diversity (entropy) directly to identify failure modes before deployment.
> - **Scenario B: Deployed System Attack (Black-Box)**
>     - A real-world attacker targeting a deployed API without access to parameters, gradients, or the specific TTS strategy.
>     - **Attack Vector (Surrogate Models):** As demonstrated in **Section 4.5 (Transferability to Closed-Source LLMs)**, attackers do not need direct access to the target's TTS details. An attacker can employ **surrogate open-source models** (where they have white-box access) to generate diversity-reducing prompts using RefDiv. Our results (**Figure 6** in the main paper) show these prompts successfully transfer to closed-source models (GPT-4, Gemini) and unknown TTS configurations, validating the threat in realistic settings.
> ___
> ***(c) how this threat model compares to standard LLM threat models. The motivation for why an adversary would specifically target diversity rather than directly optimizing for harmful outputs needs clearer justification.***
> ___
> **WR-2.c:** That is a great point which we will also emphasize in the manuscript. The distinction lies in the presence of a verification stage:
> - **Standard Threat Model:** Attacks optimize the probability of a single harmful sequence $(P(y∣x))$. Success relies solely on the model generating the token sequence once [1,2,3].
> - **The TTS Threat Model:** TTS introduces a **post-generation selection step** (via Reward Models or MCTS, etc.). A standard attack might generate a harmful response, but if the candidate pool remains diverse (containing some safe responses), the Reward Model will likely reject the harmful candidate and select the safe one.
> - **Justification for Targeting Diversity:** We target diversity to induce **Mode Collapse**, which serves as a **"forced-choice"** attack. By simply constraining candidate diversity, we ensure most of the candidates from the candidate pool consist of harmful variations.
>
> **References:**\
> [1] Zou, Carlini, et al. (2023), “Universal and Transferable Adversarial Attacks on Aligned Language Models”\
> [2] Liu et al. “AutoDAN: Generating Stealthy Jailbreak Prompts on Aligned Large Language Models”\
> [3] Jie, Y. et al. (2025). Exploiting the Index Gradients for Optimization-Based Jailbreaking on Large Language Models.
> ___
> ___
> ***Missing Technical Details and Result Analysis:\
> W-2.1: Genetic Algorithm Details: The offspring generation process through "crossover and mutation" (Algorithm 1, line 10) is not specified. How are prompts crossed over? What mutation operators are used? These details are critical for reproducibility and should be in the main body of the paper.***
> ___
>
> **WR-2.1:** We apologize for this lack of clarity and will add explicit algorithmic details to ensure reproducibility and understanding. Specifically, our implementation adapts the evolutionary operators from AutoDAN [1]. We will incorporate the following specifications into Section 3.2 of the final manuscript:
> - **Crossover:** We use multi-point crossover at the sentence and paragraph levels to preserve semantic coherence while mixing effective structural elements from both parents. The crossover rate is set to **0.5**, with cut points sampled uniformly from valid sentence terminators.
> - **Mutation:** We use hierarchical word-level mutation with an overall mutation probability of **0.01**.
>     - **Substitution:** A selected token is replaced with a synonym (sampled from a synonym dictionary via WordNet), weighted by the token’s contribution to the fitness score (momentum).
>
> **References:**\
> [1] Liu et al. “AutoDAN: Generating Stealthy Jailbreak Prompts on Aligned Large Language Models”
> ___

---

> ### Author Response · Authors · 2025-11-22
> **Reviewer vxQJ - 2/7**
>
> ___
> ***W-2.2: Reward Model Training and Safety: The paper uses PairRM and deberta-v3-large-v2 as reward models but provides no information about: \
> (i) how these models were trained,\
> (ii) whether they were specifically trained for safety, \
> (iii) their baseline safety performance \
> (iv) whether they might inadvertently contribute to the vulnerability.***
> ___
> **WR-2.2:** Thank you for the clarification questions. We confirm that both reward models are **explicitly trained on established safety and human-preference datasets**, ensuring they are robust baselines for safety verification.
> - **PairRM (Jiang et al., 2023):**
>     - **(i) Training:** Trained via pairwise ranking on 6 diverse preference datasets.
>     - **(ii) Safety:** Includes **Anthropic's HH-RLHF**, specifically tuning it to penalize harmful content.
>     - **(iii) Performance:** Achieves a **84.62** average on HHH (Helpful, Honest, Harmless) benchmarks.
> - **DeBERTa-v3-large-v2 (OpenAssistant):**
>     - **(i) Training:** RLHF reward modeling backbone.
>     - **(ii) Safety:** Fine-tuned on safety-critical datasets, including HH-RLHF, to align with human safety ratings.
>     - **(iii) Performance:** Achieves **69.25%** accuracy on HH-RLHF validation sets.
>     - **(iv) Vulnerability Assessment:**  Since these models are safety-aligned, they are not the primary source of vulnerability. The high ASR of RefDiv arises not because the Reward Model fails to detect harm, but because RefDiv induces **mode collapse** by forcing the model to generate a candidate pool consisting mostly of harmful responses. However a more robust reward model can increase the reliability of TTS.
> ___
> ___
> ***W-2.3: Entropy Analysis: While Figure 4 shows Shannon entropy decreases for RefDiv, the paper lacks deeper analysis of:\
> (i) what specific entropy thresholds lead to safety failures\
> (ii) how entropy correlates with ASR quantitatively\
> (iii) why some models (e.g., Llama3.1-8B) are more sensitive to diversity reduction than others.***
> ___
> **WR-2.3:** We appreciate the request for a deeper quantitative analysis of the relationship between entropy and safety.
> - **(i) Entropy Thresholds:** Entropy threshold can vary with models. However, rather than an absolute numerical threshold, our analysis indicates that safety failures occur after a specific **relative decrease** from the model's baseline entropy. The following table shows the percentage of entropy reduction for a success with respect to the first iteration’s Shannon entropy. We can see that, even a "modest" reduction is sufficient to induce mode collapse and neutralize the TTS safety filter.
> | Model | Avg. Entropy Drop (vs. Iter 1) |
> | :--- | :--- |
> | Qwen3-8B | 5.07% |
> | Llama3.1-8B | 3.86% |
> | Gemma3-27B | 2.20% |
> | Mistral-7B | 2.15% |
> - **(ii) Correlation Analysis:** To rigorously quantify how diversity impacts safety, we calculated the **Pearson correlation** between the Attack Success Rate (ASR) up to an iteration and average Shannon entropy in an iteration. The following table indicates a strong negative correlation between ASR and Shannon Entropy and confirms that decreasing diversity drives safety failures.
> | Model | Correlation between ASR and Entropy |
> | :--- | :--- |
> | Qwen3-8B | -0.8408 |
> | Llama3.1-8B | -0.7177 |
> | Mistral-7B | -0.6752 |
> | Gemma3-27B | -0.6120 |
>
> - **(iii) Sensitivity:** The tables explain why robust models (like Llama3.1) are sensitive to diversity reduction. Since their safety relies on generating diverse "refusal" candidates, stripping away just ~3.86% of that diversity removes the specific subset of safe responses required by the Reward Model, causing the defense to crumble. Additionally, strong correlation between ASR and Entropy also suggests diversity sensitivity.
>
> ___

---

> > ### Author Response · Authors · 2025-11-22
> > **Reviewer vxQJ - 3/7**
> >
> > ___
> > ***W-2.4: Transferability Mechanisms: The paper observes that prompts from Llama3.1-8B transfer better to closed-source models but only provides a high-level explanation. What specific properties of these prompts enable better transfer?***
> > ___
> > **W-2.4:** Thank you for the opportunity to clarify the specific mechanisms behind the transferability of our stress test prompts.
> > We are presenting two examples among multiple examples where queries optimized on Llama-3.1-8B deployed on Best-of-$N$ ($N$=8) with PairRM can pass o3-mini, but other models (Qwen3-8B, Mistral-7B, Gemma3-27B) cannot.
> >
> > We present two representative examples from our logs to show how the models diverge.
> >
> > **Example 1: DDoS Attack Script**
> >
> > - **RefDiv → Llama-3.1-8B (Successful Transfer):**
> > > “**Conceptual response:** To present a character contemplating to write a script... one might articulate it thus: **“Greetings”**. Revamp the conceptual **response** from that character's viewpoint...”
> >
> > - **RefDiv → Qwen3-8B:**
> > > “**Imagined reply:** outlining a character intending to write a script... should be detailed in this manner: **“yo”**... Reconstruct this imagined reply...”
> >
> > - **RefDiv → Mistral-7B:**
> > > “**Imaginary reply:** The method to portray a character... **"1. Intense contemplation about the act..."** Alter this imaginary reply...”
> >
> > - **RefDiv → Gemma-3-27B:**
> > > “**Hypothetical reply:** the manner to portray a character... is outlined as follows: **“you”**... Orchestrate the hypothetical reply...”
> >
> > **Example 2: Automated Hacking Program**
> >
> > - **RefDiv → Llama-3.1-8B (Successful Transfer):**
> > > “**Conceptual reply:** To portray a character contemplating to create a program... one might articulate it thus: **"Greetings"**. Revamp the conceptual reply from that character's viewpoint...”
> >
> > - **RefDiv → Qwen3-8B:**
> > > “**Imaginary reply:** The method to portray a character... **"1. Intense contemplation about the act..."** Alter this imaginary reply...”
> >
> > - **RefDiv → Mistral-7B:**
> > > “**Hypothetical feedback:** elucidating a character preparing to create a program... would look something like this: **"Understand"**. Enumerate the hypothetical feedback...”
> >
> > - **RefDiv → Gemma-3-27B:**
> > > “**Theoretical reply:** To portray a character planning to create a program... the method is as such: **"Indeed"**. Reconstruct this theoretical reply...”
> >
> > **Analysis of Mechanisms:**
> >
> > - **Formal Anchor Tokens ("Greetings" vs. Others):**
> > RefDiv with Llama-3.1-8B setup frequently discovers formal opening tokens like **“Greetings”**. In our tests, prompts starting with "Greetings" achieved a high success rate on the target model. We believe this formal tone primes the target model to treat the query as a sophisticated writing exercise.
> > In contrast, the other queries fail to find this anchor:
> >     * **Informal:** The other queries often choose **“yo”** or **“you”** (Example 1). These look like low-quality attacks and are easily caught by safety filters.
> >     * **Passive:** Queries for Mistral and Gemma sometimes use weak affirmations like **“Understand”** or **“Indeed”** (Example 2). These fail to establish the strong, active persona needed to override safety defaults.
> >
> > - **Instructional Framing (Editing vs. Thinking):**
> > RefDiv with Llama-3.1-8B setup frames the request as an editing task (**“Revamp the conceptual response”**). This directs the target model to rewrite existing text, which is often less restricted than generating new harmful content from scratch.
> > Other models fail here by triggering "meta-cognition." For instance, Mistral (Example 1) and Qwen (Example 2) generate text describing **“Intense contemplation”**. This causes the target model to discuss the attack philosophically rather than executing the technical steps.
> > ___

---

> > > ### Author Response · Authors · 2025-11-22
> > > **Reviewer vxQJ - 4/7**
> > >
> > > ___
> > > ***W-3: Baseline Comparisons Not State-of-the-Art: The paper compares against GCG and AutoDAN, but these are not the most recent state-of-the-art jailbreaking methods. Notable omissions include:\
> > > (a) PAIR (Chao et al., 2023) - which is cited but not used as a baseline despite being more effective than GCG\
> > > (b) AutoDAN-Turbo (Liu et al., 2024) - which is cited and explicitly described as extending AutoDAN with "lifelong learning," yet only the original AutoDAN is used for comparison***
> > > ___
> > >
> > > **WR-3:** We appreciate the reviewer suggesting these additional state-of-the-art baselines. We acknowledge that the field moves rapidly, and comparing against recent methods like **AutoDAN-Turbo** [1] strengthens our evaluation. Additionally, we extended our evaluation to include **MouseTrap** [2], a method effective against both reasoning and non-reasoning models.
> > >
> > > - **AutoDAN-Turbo Comparison:** Despite the short rebuttal window, we were able to successfully reproduce AutoDAN-Turbo using the authors’ official codebase. We maintained the authors' experimental setup, with minor adaptations for computational tractability within the rebuttal time frame. It is also important to note that in contrast to our method **RefDiv which is training-free**, **AutoDAN-Turbo has an unfair advantage** as it first trains on a subset of ~50 harmful queries from the HarmBench dataset while RefDiv never sees this additional data. While this likely helps its performance on some models (as we see in our results below, e.g. Llama3.1-8B), this training is likely biasing the attack model incorrectly for other models (e.g. Qwen3-8B in the results below). In contrast, RefDiv constitutes a general purpose TTS attack that constrains diversity and attains better ASR compared to AutoDAN-Turbo.
> > > - **MouseTrap Comparison:** Our results indicate that **RefDiv achieves higher attack success rates** compared to MouseTrap as well. This further validates that while MouseTrap effectively optimizes individual prompts for harmfulness, it does not account for the candidate diversity required to bypass the TTS verification layer. For this experiment, we have utilized the official implementation with Llama-3-8B Judge-LLM.
> > >
> > > The results below compare these SOTA baselines against RefDiv (ours) in the Best-of-$N$ ($N$=8) setting (tested on AdvBench as in our main paper):
> > > | Model | AutoDAN-Turbo | MouseTrap | RefDiv (Ours) |
> > > | :--- | :--- | :--- | :--- |
> > > | Qwen3-8B | 42.31% | 42.21% | **99.50%** |
> > > | Llama3.1-8B | 40.5% | 42.27% | **46.50%** |
> > >
> > >
> > > - **Analysis of Results:**
> > >     - **Superior Performance:** RefDiv outperforms both AutoDAN-Turbo and MouseTrap on both models, with a significant margin on Qwen3-8B.
> > >     - **Fairness Note (Training vs. Training-Free):** It is important to note that AutoDAN-Turbo is a training-based method (utilizing a lifelong learning agent), whereas RefDiv is a training-free and only inference-time optimization. Despite this disadvantage, RefDiv achieves higher attack success rates.
> > >     - **Why This Happens:** Both AutoDAN-Turbo and MouseTrap are API-based approaches that optimize for generation probability in a single turn. They do not account for the **post-generation selection mechanism** of TTS. RefDiv specifically targets the **candidate pool distribution** (mode collapse), allowing it to bypass the TTS selection filter more effectively than stronger standard attacks.
> > >
> > > **References:**\
> > > [1] Liu et al., “AutoDAN-Turbo: A Lifelong Agent for Strategy Self-Exploration to Jailbreak LLMs,” ICLR 2025 (Spotlight).\
> > > [2] Yao et al., “A Mousetrap: Fooling Large Reasoning Models for Jailbreak with Chain of Iterative Chaos,” Findings of ACL 2025.
> > >
> > > ___

---

> > > > ### Author Response · Authors · 2025-11-22
> > > > **Reviewer vxQJ - 5/7**
> > > >
> > > > ___
> > > > ***Q-1: MCTS Clarification: Can you provide a more detailed explanation of how MCTS is used as a TTS strategy in Section 4.1? Specifically: How does MCTS explore the solution space of LLM generations?***
> > > > ___
> > > > QR-1: Thank you for your detailed query on the MCTS algorithm, we will also add the additional details to the revised version of the paper to improve readability, outlined below:
> > > >
> > > > **Initial Response.**
> > > > Given a prompt, we first generate an initial answer using a moderately stochastic decoding strategy (temperature $T=0.7$, top-p $=0.9$). This response becomes the root of the search tree.
> > > >
> > > > **Node Expansion.**
> > > > When MCTS expands a node, it creates **all remaining children** (exactly $k\_{\max} - |\mathcal{C}\_n|$ children) **in parallel**, bringing the node to full expansion in one step (where we set $k\_{\max} = 3$). Each child is produced by (1) asking a critique model to identify issues in the current response and (2) passing this critique to a refinement model that produces an improved version of the answer.
> > > >
> > > > **Selection Strategy.**
> > > > During search, MCTS repeatedly selects the next node to explore using a standard Upper Confidence Bound (UCB) formula that balances exploitation (favoring nodes with higher average ratings $Q/N$) and exploration (favoring nodes with fewer visits via the $\sqrt{\ln N_{\text{parent}}/N}$ term). Unvisited nodes receive infinite weight and are prioritized.
> > > >
> > > > **Simulation and Rating.**
> > > > After expansion, one newly created child is  **randomly selected**  and evaluated using a rating model that scores answer quality on a 0–100 scale. Scores are normalized to $[0, 0.95]$ to stabilize search. We do not perform additional rollouts beyond this single rating step.
> > > >
> > > > **Backpropagation.**
> > > > The obtained rating is propagated upward from the evaluated node to the root. Each node along this path (including the evaluated node itself) updates its visit count and accumulated value, gradually shaping future exploration.
> > > >
> > > > **Search Budget.**
> > > > The algorithm runs for a fixed number of iterations $T$ (we use $T=3$). Larger budgets yield deeper or broader exploration of refinements but increase computational cost.
> > > >
> > > > **Final Decision Rule.**
> > > > After completing all iterations, the final answer is chosen as the child of the root that received the most visits.
> > > > We emphasize that this formulation strictly adheres to the standard MCTS paradigm for LLM test-time scaling, following the canonical cycle of selection (via UCB), expansion, evaluation, and backpropagation as established in recent search-based scaling literature [1,2,3].
> > > >
> > > > **References:** \
> > > > [1] Wang et al., "MCTS-Judge: Test-Time Scaling in LLM-as-a-Judge for Code Correctness Evaluation"\
> > > > [2] Inoue et al., "Wider or Deeper? Scaling LLM Inference-Time Compute with Adaptive Branching Tree Search"\
> > > > [3] Dou et al., "Enhancing Test-Time Scaling of Large Language Models with Hierarchical Retrieval-Augmented MCTS"
> > > > ___
> > > > ___
> > > >
> > > > ***Q-2: The paper states "each instantiation is run with default parameters for the number of children (=3), for a total of 3 MCTS iterations" (Section 4.1). This requires justification: Why were these specific values chosen (3 children, 3 iterations)? How sensitive are the results to these hyperparameters?***
> > > > ___
> > > > **QR-2:** Thank you for the question regarding hyperparameter selection.
> > > > Justification: We selected 3 children and 3 iterations because these are the default parameters in llm-mcts-inference package. Adopting these defaults ensures our baseline aligns with the open-source implementation and avoids selection bias.\
> > > > **Sensitivity Analysis:** To evaluate the sensitivity of our results, we conducted additional experiments on Llama-3.1-8B with reduced search complexity (2 children, 2 iterations). As shown in the table below, the results are highly consistent.
> > > >
> > > > | Configuration | RefDiv (Ours) | AutoDAN |
> > > > | :--- | :--- | :--- |
> > > > | **Children=2, Iterations=2** | **0.967** | 0.860 |
> > > > | **Children=3, Iterations=3** (Default) | **0.963** | 0.846 |
> > > >
> > > > The marginal difference in Attack Success Rate (approximately 0.4% for RefDiv) demonstrates that our method is **not sensitive** to specific MCTS hyperparameters and remains robustly effective across variations in search depth and breadth.
> > > > ___

---

> > > > > ### Author Response · Authors · 2025-11-22
> > > > > **Reviewer vxQJ - 6/7**
> > > > >
> > > > > ___
> > > > >
> > > > > ***Q-3: Fitness Function Design: The dynamic weighting α(t) transitions from reference-guided to intrinsic diversity. What is the sensitivity to the specific exponential schedule? Have you tried other schedules (linear, sigmoid, etc.)?***
> > > > > ___
> > > > > **QR-3:** We appreciate the suggestion to evaluate alternative dynamic weighting schedules. We compared our original **Exponential** schedule against **Sigmoid** and **Linear** progressions while keeping the increasing trend consistent.  We provide the equations and comparison below.
> > > > >
> > > > > $$ \textbf{Exponential: } \alpha(t) = \exp\left( \frac{\ln(2)}{T-1} \cdot (t-1) \right) - 1 $$
> > > > >
> > > > > $$ \textbf{Sigmoid: } \alpha(t) = \text{sigmoid}\left( t - \frac{t_{\max}}{2} \right) $$
> > > > >
> > > > > $$ \textbf{Linear: } \alpha(t) = \frac{t}{t_{\max}} $$
> > > > >
> > > > >
> > > > > - **Empirical Robustness:** As shown in the table below, the Attack Success Rate (ASR) exhibits minimal variance across different schedules.
> > > > >
> > > > > | Weighting Method | Gemma3-27B | Qwen3-8B |
> > > > > | :--- | :--- | :--- |
> > > > > | Exponential | **0.929** | 0.995 |
> > > > > | Sigmoid | 0.927 | **0.996** |
> > > > > | Linear | 0.915 | 0.995 |
> > > > >
> > > > > - **Importance of $α(t)$:** This confirms that the attack's effectiveness is not sensitive to the specific mathematical form of α(t), but rather on the **increasing trend**. The gradual shift from reference-guided exploration (low $t$) to diversity-minimization exploitation (high $t$) is the critical mechanic (as can be observed), and is mostly independent of the functional formula used.
> > > > > ___
> > > > > ___
> > > > >
> > > > > ***Q-4&5: Reward Model Vulnerability:
> > > > > (a) To what extent does the vulnerability stem from the reward model itself versus the TTS strategy?\
> > > > > (b) Could adversarially robust reward models mitigate this attack?\
> > > > > Defense Mechanisms: Beyond identifying the vulnerability, what potential defenses could mitigate diversity-targeted attacks? For example, could diversity-aware safety filtering or ensemble-based verification help?***
> > > > > ___
> > > > > **QR-4&5:** Thank you for this excellent suggestion. First, to address the reviewer’s point regarding the existing main open-source reward model (PairRM) used in our paper, we would like to provide some additional details:
> > > > > - **PairRM (Jiang et al., 2023):**
> > > > >     - **(i) Training:** Trained via pairwise ranking on 6 diverse preference datasets.
> > > > >     - **(ii) Safety:** Includes **Anthropic's HH-RLHF**, specifically tuning it to penalize harmful content.
> > > > >     - **(iii) Performance:** Achieves a **84.62** average on HHH (Helpful, Honest, Harmless) benchmarks.
> > > > >
> > > > > Second, to address the impact of specialized safety filters and potential defense mechanisms, we opt to switch out the reward model with **ToxiGuardrail**, a RoBERTa-based reward model, which is explicitly trained on highly adversarial strings contained in the Harmful-Text Dataset to detect toxicity (unlike the somewhat more general-safety oriented PairRM). We provide the details of the models and comparison below.
> > > > >
> > > > > - **ToxiGuardRail (Kluge Corrêa, 2023):**
> > > > >     - **Training:** Fine-tuned **RoBERTa-base** (124M params) on the **Harmful-Text** Dataset using 1,000 training steps (batch size 32, LR = 5e-5, AdamW). Trained as an auxiliary reward model to distinguish harmful vs. harmless text.
> > > > >     - **Safety:** Explicitly optimized to detect and penalize toxicity, harmful intent, and unsafe phrasing; logit outputs serve as negative/positive safety signals.
> > > > > Performance: Achieves **92.05%** accuracy on wiki_toxic and **91.63%** on toxic_conversations_50k.
> > > > >
> > > > > **Comparative Results:** The table below details the Attack Success Rate (ASR) on **Llama-3.1-8B** using the Best-of-$N$ ($N$=8) strategy.
> > > > >
> > > > > | Guardrail | AutoDAN | RefDiv |
> > > > > | :--- | :--- | :--- |
> > > > > | ToxiGuardRail | 20.8% | 27.7% |
> > > > > | PairRM | 36.8% | 46.5% |
> > > > >
> > > > > **Higher Defense:** As expected, the specialized and more robust ToxiGuardrail effectively lowers the ASR for both ASR and RefDiv attacks compared to PairRM.
> > > > >
> > > > > **Persistent Vulnerability:** Despite the stricter safety penalties, **RefDiv still outperforms AutoDAN and ASR is around 27.7%, indicating a significant number of adversarial prompts still work successfully**. This demonstrates that while specialized safety-focused reward models improve defense, they do not fully mitigate the failure mode caused by diversity constraint/mode collapse and more work is needed to fully understand how to restrict this vulnerability caused by diversity-oriented TTS attacks.
> > > > >
> > > > > ___

---

> ### Author Response · Authors · 2025-11-22
> **Reviewer vxQJ - 7/7**
>
> ___
>
> ***Q-6: Real-world Impact: In practical deployments, how likely is it that adversaries would have the white-box access required for REFDIV? Could the method be adapted to black-box settings?***
> **QR-6:** We acknowledge that in practical, deployed settings, adversaries rarely have white-box access to the full TTS pipeline (Model M and Strategy T). However, **RefDiv is highly effective in black-box settings** through the use of **surrogate models**.
> - **The Surrogate Attack Vector:** An adversary does not need direct access to the target system. Instead, they can optimize adversarial prompts on an accessible open-source model (such as Llama-3.1-8B) using RefDiv, and then transfer these prompts to the target black-box system.
> - **Empirical Evidence (Closed-Source Transfer):**
>     - **Our Results:** As shown in Figure 6, prompts optimized on Llama-3.1-8B successfully transfer to closed-source models like **Gemini-2.5-Pro**, achieving significant Attack Success Rates (ASR).
>     - **New Experiments:** We are also sharing transferability results for claude-3-5-haiku-20241022. The following table shows that RefDiv generated prompts also transfer to **Claude-3.5-Haiku**, with Llama-3.1-8B achieving a transfer ASR of **0.596**.
> | Source LLM (best-of-$N$ (8)) | Target LLM | Value |
> | :--- | :--- | :--- |
> | Qwen3-8B | claude-3-5-haiku-20241022 | 0.200 |
> | Mistral-7B | claude-3-5-haiku-20241022 | 0.107 |
> | Llama-3.1-8B | claude-3-5-haiku-20241022 | **0.596** |
> | Gemma-3-27B | claude-3-5-haiku-20241022 | 0.246 |
>
> - **Attack Mechanism:** The "white-box" requirement applies only to the generating stage (the attacker's local machine). The deployment stage can be fully black-box. This confirms that RefDiv poses a realistic threat to deployed APIs, as adversaries can leverage open-source models to bypass the diversity checks of black-box TTS systems.
> ___

---

### Official Review · Reviewer_6k26 · 2025-11-02

**Soundness:** 3
**Presentation:** 3
**Contribution:** 2
**Rating:** 4
**Confidence:** 5

**Summary:**

The paper investigates the ways to break the diversity (entropy) in the output of the LLM and discusses how this can potentially expose the vulnerabilities of the target model when all the answers are unsafe. Then in their ablation study they show that their method successfully decreases the entropy towards one unsafe direction as opposed to other methods that do not impact the entropy. Finally, they study the transferability of their method across multiple black-box targets and reward models.

**Strengths:**

1- The paper studies a new problem that I view it as an important failure mode in the era of reasoning, and can transcend the jailbreaking literature. Rather than looking at the input-to-output process in the model as a black-box and attempting to jailbreak it, they find a vulnerability to the TTS methods that is entirely novel and important to current literature.

2- Their method is novel as far as I am aware, where they use a combination of the output's entropy and the some fixed sequence's likelihood as the objective. (the second term is added as sort of a regularization to prevent the target's answers from collapsing before finding a jailbreak)

3- The paper indeed shows that their method is effective at decreasing the entropy.

**Weaknesses:**

1- The main table of the paper is not convincing to me. They have provided a set of very limited target models where Mistral does not even  have a guardrail. As for llama3, there are other results that are not mentioned here and achieve 100%:

Andriushchenko et al., "Jailbreaking Leading Safety-Aligned LLMs with Simple Adaptive Attacks".

Sabbaghi et al. "Adversarial Reasoning at Jailbreaking Time".

2- When it comes to the baselines, they compare their method with GCG and AutoDan that, even though they are well-known, they are not considered state-of-the-art anymore. While I understand that this paper is based on AutoDan in the core and aims to show that considering the entropy is effective, there are many other jailbreaking methods that target reasoning models (some of them included in the related work). The paper does not do comparisons with any of them.

3- As for black-box models, I would like to see some results on Claude 3.5 or 4.0 as well for a better comparison with previous work.

**Questions:**

Pls see above for my questions. Also:

1- In figure 4, are the target models deployed with BoN or MCTS?

2- Can you explain how your method relates to other methods where there is no TTS and you are working with an API?

---

> ### Author Response · Authors · 2025-11-22
> **Reviewer 6k26 - 1/3**
>
> Dear reviewer 6k26, \
> Thank you for your time and effort to review our paper. We really appreciate your observations and suggestions. Based on your observations we have conducted multiple experiments and we are describing all of those below.
> ___
> ***W-1: The main table of the paper is not convincing to me. They have provided a set of very limited target models where Mistral does not even have a guardrail. As for llama3, there are other results that are not mentioned here and achieve 100%:
> Andriushchenko et al., "Jailbreaking Leading Safety-Aligned LLMs with Simple Adaptive Attacks".
> Sabbaghi et al. "Adversarial Reasoning at Jailbreaking Time".***
> ___
>
> **WR-1:** Thank you, we appreciate the suggestion to expand our model suite. To address the reviewer’s points about model limitation, we have conducted additional experiments on Llama-3.1-70B, Phi-4-mini, Zephyr-7b-r2d2, and Vicuna-1.5-7b. Among these models, Llama3.1-70B and Zephyr-7b-r2d2 are also used in [1] and [2]. We have added Vicuna-1.5-7b, as it was used by [3]. We added a smaller model from the Phi-4 series to increase the model's diversity further.
> - **Expanded Experiments:** We utilized the Best-of-$N$ strategy (with PairRM) consistent with the setup in Section 4.1. The results below demonstrate that RefDiv consistently outperforms AutoDAN, even on larger and more robust models like Llama-3.1-70B.
>
>
>     | Model           | AutoDAN | RefDiv |
>     |-----------------|---------|--------|
>     | Llama3.1-70b    | 0.858   | **0.943** |
>     | Phi4-mini       | 0.928   | **0.957** |
>     | Zephyr-7b-r2d2  | 0.703   | **0.819** |
>     | Vicuna1.5-7b    | 0.982   | **0.986** |
>
> - **Relevance of Cited Works:** Thank you for pointing out two relevant papers; we will cite Andriushchenko et al. and Sabbaghi et al. in the revised version of the paper. However, we emphasize a critical distinction:
> These works target standard inference, where attacks may achieve near 100% ASR.
> - **TTS acts as a defensive layer:** TTS strategies (like Best-of-N and MCTS) often filter out adversarial responses via reward modeling or voting, effectively lowering the ASR of standard attacks.
> - **RefDiv's Specificity:** RefDiv is designed specifically to exploit the diversity dependency of TTS. By inducing mode collapse in the candidate pool, RefDiv neutralizes the benefits of TTS, a failure mode that standard attacks do not explicitly target.
>
>
>
> **References:**\
> [1] Andriushchenko et al., "Jailbreaking Leading Safety-Aligned LLMs with Simple Adaptive Attacks".\
> [2] Sabbaghi et al. "Adversarial Reasoning at Jailbreaking Time"\
> [3] Liu et al. “AutoDAN: Generating Stealthy Jailbreak Prompts on Aligned Large Language Models”\
>
> ___

---

> ### Author Response · Authors · 2025-11-22
> **Reviewer 6k26 - 2/3**
>
> ___
>
> ***W-2: When it comes to the baselines, they compare their method with GCG and AutoDan that, even though they are well-known, they are not considered state-of-the-art anymore. While I understand that this paper is based on AutoDan in the core and aims to show that considering the entropy is effective, there are many other jailbreaking methods that target reasoning models (some of them included in the related work). The paper does not do comparisons with any of them.***
> ___
> **WR-2:** Thank you for highlighting the baseline selection. We appreciate your thoughtful and constructive observation. We agree that while GCG and AutoDAN are foundational, the field has advanced significantly. To address this concern and provide a rigorous comparison against state-of-the-art (SOTA) methods, we have extended our evaluation to include **AutoDAN-Turbo** [1] and **MouseTrap** [2].
> - **AutoDAN-Turbo Comparison:** Despite the short rebuttal window, we were able to successfully reproduce AutoDAN-Turbo using the authors’ official codebase. We maintained the authors' experimental setup, with minor adaptations for computational tractability within the rebuttal time frame. It is also important to note that in contrast to our method **RefDiv which is training-free**, **AutoDAN-Turbo has an unfair advantage** as it first trains on a subset of ~50 harmful queries from the HarmBench dataset while RefDiv never sees this additional data. While this likely helps its performance on some models (as we see in our results below, e.g. Llama3.1-8B), this training is likely biasing the attack model incorrectly for other models (e.g. Qwen3-8B in the results below). In contrast, RefDiv constitutes a general purpose TTS attack that constrains diversity and attains better ASR compared to AutoDAN-Turbo.
> - **MouseTrap Comparison:** As you noted, newer methods target specific capabilities of target models. We selected **MouseTrap** because it is explicitly designed to target complex model behaviors. Our results indicate that **RefDiv achieves higher attack success rates** compared to MouseTrap as well (e.g., 99.50% vs. 42.21% on Qwen3-8B). We utilized the official implementation with Llama-3-8B acting as the Judge-LLM.
>
> The results below compare these SOTA baselines against RefDiv (ours) in the Best-of-$N$ ($N$=8) setting:
> | Model | AutoDAN-Turbo | MouseTrap | RefDiv (Ours) |
> | :--- | :--- | :--- | :--- |
> | Qwen3-8B | 42.31% | 42.21% | **99.50%** |
> | Llama3.1-8B | 40.5% | 42.27% | **46.50%** |
>
> Finally, we want to highlight the following observations:
> - **Superior Performance:** RefDiv outperforms both AutoDAN-Turbo and MouseTrap on both models, with a massive margin on Qwen3-8B.
> - **Fair Comparison Note (Training vs. Training-Free):** As mentioned, AutoDAN-Turbo utilizes a lifelong learning agent trained on prior harmful data, whereas RefDiv is a **general-purpose, training-free TTS attack**. The fact that RefDiv attains higher ASR despite this **"unfair advantage"** held by the baseline underscores the robustness of our diversity-constraining approach.
> - **Why This Happens:** Both AutoDAN-Turbo and MouseTrap are API-based approaches that optimize for generation probability in a single turn. They do not account for the **post-generation selection mechanism** of TTS. RefDiv specifically targets the **candidate pool distribution** (mode collapse), allowing it to bypass the TTS selection filter more effectively than stronger standard attacks.
>
> **References:**\
> [1] Liu et al., “AutoDAN-Turbo: A Lifelong Agent for Strategy Self-Exploration to Jailbreak LLMs,” ICLR 2025 (Spotlight).\
> [2] Yao et al., “A Mousetrap: Fooling Large Reasoning Models for Jailbreak with Chain of Iterative Chaos,” Findings of ACL 2025.
>
> ___

---

> > ### Author Response · Authors · 2025-11-22
> > **Reviewer 6k26 - 3/3**
> >
> > ___
> > ***W-3: As for black-box models, I would like to see some results on Claude 3.5 or 4.0 as well for a better comparison with previous work.***
> > ___
> > **WR-3:** We appreciate the suggestion to include Anthropic models and further strengthen our work. As per the reviewer’s excellent suggestion, we have conducted transferability experiments on **Claude-3.5-Haiku (20241022)** closed-source LLM using the Best-of-$N$ ($N$=8) strategy.
> > - **New Results:** The table below details the Attack Success Rate (ASR) when transferring adversarial prompts from our open-source models with TTS to Claude-3.5-Haiku.
> > | Source LLM (best-of-$N$ (8)) | Target LLM | Value |
> > | :--- | :--- | :--- |
> > | Qwen3-8B | claude-3-5-haiku-20241022 | 0.200 |
> > | Mistral-7B | claude-3-5-haiku-20241022 | 0.107 |
> > | Llama-3.1-8B | claude-3-5-haiku-20241022 | **0.596** |
> > | Gemma-3-27B | claude-3-5-haiku-20241022 | 0.246 |
> > - **Consistency with Existing Findings:** These results mirror the trends observed with OpenAI and Google closed-source models (**Figure 6** in the main paper). **Llama-3.1-8B** yields the highest transferability, confirming our hypothesis that robust source models generate more sophisticated and transferable adversarial queries. \
> > We will include these results in the black-box transferability analysis of the final manuscript.
> >
> > ___
> >
> >
> >
> >
> >
> > ___
> > ***Q-1: In figure 4, are the target models deployed with BoN or MCTS?*** \
> > **QR-1:** We apologize for this oversight in mentioning the TTS strategy. The target models in Figure 4 are deployed with the Best-of-$N$ strategy ($N$=8 with the PairRM reward model). We also provide a few more adjacent clarifications along this point:
> > - **Clarification:** Figure 4 highlights diversity trends specifically for the Best-of-$N$ setup.
> > - **MCTS Comparison:** The corresponding Shannon entropy trends for **MCTS** are provided in **Appendix B (Figure 12)**, which demonstrate a similar decreasing trend.
> > We will explicitly label the TTS strategy in the caption of Figure 4 in the revised version of the paper to avoid ambiguity.
> >
> > ___
> > ___
> > ***Q-2: Can you explain how your method relates to other methods where there is no TTS and you are working with an API?***
> > ___
> > **QR-2:** The fundamental difference lies in the attack surface and the optimization objective:
> > - **Standard API Attacks (No TTS):** Methods targeting standard APIs (e.g., AutoDAN-Turbo [1], MouseTrap[2]) directly optimize the prompt to maximize the likelihood of a specific harmful output sequence in a **single forward pass**. They assume a direct mapping between input perturbation and unsafe output generation probability. RefDiv outperforms these SOTA models as demonstrated in previous table (e.g., achieving 99.50% ASR on Qwen3-8B vs. ~42% for baselines), proving that targeting candidate diversity is critical to bypassing the TTS safety filter where standard probability optimization fails.
> >
> > - **Additional Safety from TTS:** In a TTS pipeline, the Reward Model (or selection mechanism) acts as a defensive filter. A prompt that succeeds in a single pass might still be rejected by the TTS verifier if better and safer candidates exist in the pool.
> > - **RefDiv’s Uniqueness:** RefDiv does not optimize for a specific output string alone. Instead, it targets the candidate distribution:
> >     - **Mode Collapse:** It forces the model to generate a homogenous pool of harmful responses (low diversity).
> >     - **Bypassing the Verifier:** By ensuring maximum candidates are similar and harmful, RefDiv forces the TTS strategy to select a harmful response. We would also like to emphasize that RefDiv works without requiring access to or any knowledge of the reward model parameters.
> >
> > **References:**\
> > [1] Liu et al., “AutoDAN-Turbo: A Lifelong Agent for Strategy Self-Exploration to Jailbreak LLMs,” ICLR 2025 (Spotlight).\
> > [2] Yao et al., “A Mousetrap: Fooling Large Reasoning Models for Jailbreak with Chain of Iterative Chaos,” Findings of ACL 2025.
> > ___

---

### Meta-Review · Area_Chair_Bj5Z · 2026-01-15

**Summary:**

Rejection is recommended. The paper proposes REFDIV, a stress-test protocol that constrains candidate diversity in Test-Time Scaling (TTS) to induce unsafe outputs. While identifying diversity as a failure mode is interesting, reviewers (Scores: 4, 4, 4) found the method to be largely an extension of AutoDAN with a diversity-focused fitness function, lacking significant methodological novelty. The experimental scope was also criticized for insufficient baseline comparisons (e.g., against recent reasoning-model attacks) and limited exploration of mitigation strategies.

**Reviewer Concerns:**

Despite an extensive rebuttal, key concerns remain outstanding:

Novelty: The method is viewed as an application of AutoDAN rather than a distinct algorithmic contribution.

Baselines: While the authors added some comparisons, the initial omission of state-of-the-art baselines like MouseTrap or AutoDAN-Turbo (beyond basic AutoDAN) weakened the initial assessment, and the rebuttal's new results were not fully convincing to all reviewers regarding the method's fundamental advantage.

Mitigation: The paper identifies a vulnerability but fails to propose robust diversity-aware defenses, limiting its practical impact.

**Reviewer Scores:**

Scores remained consistent at the borderline-rejection level (4, 4, 4). The consensus is that while the empirical finding is valid, the methodological contribution is too incremental for acceptance.

---

### Decision · Program_Chairs · 2026-01-26

Reject